# E3Bind: An End-to-End Equivariant Network for Protein-Ligand Docking

**Yangtian Zhang**[1,2*]**, Huiyu Cai**[1,2*]**, Chence Shi**[1,2]**, Jian Tang**[1,3,4]
[1]Mila - Québec AI Institute, Canada [2]Université de Montréal, Canada
[3]HEC Montréal, Canada [4]CIFAR AI Research Chair
{yangtian.zhang, huiyu.cai, chence.shi}@umontreal.ca
jian.tang@hec.ca

## Abstract

*In silico* prediction of the ligand binding pose to a given protein target is a crucial but challenging task in drug discovery. This work focuses on flexible blind self-docking, where we aim to predict the positions, orientations and conformations of docked molecules. Traditional physics-based methods usually suffer from inaccurate scoring functions and high inference costs. Recently, data-driven methods based on deep learning techniques are attracting growing interest thanks to their efficiency during inference and promising performance. These methods usually either adopt a two-stage approach by first predicting the distances between proteins and ligands and then generating the final coordinates based on the predicted distances, or directly predicting the global roto-translation of ligands. In this paper, we take a different route. Inspired by the resounding success of AlphaFold2 for protein structure prediction, we propose E3Bind, an end-to-end equivariant network that iteratively updates the ligand pose. E3Bind models the protein-ligand interaction through careful consideration of the geometric constraints in docking and the local context of the binding site. Experiments on standard benchmark datasets demonstrate the superior performance of our end-to-end trainable model compared to traditional and recently-proposed deep learning methods.

## 1 Introduction

For nearly a century, small molecules, or organic compounds with small molecular weight, have been the major weapon of the pharmaceutical industry. They take effect by ligating (binding) to their target, usually a protein, to alter the molecular pathways of diseases. The structure of the protein-ligand interface holds the key to understanding the potency, mechanisms and potential side effects of small molecule drugs. Despite huge efforts made for protein-ligand complex structure determination, there are by far only some $10^4$ protein-ligand complex structures available in the protein data bank (PDB) (Berman et al., 2000), which dwarfs in front of the enormous combinatorial space of possible complexes between $10^{60}$ drug-like molecules (Hert et al., 2009; Reymond & Awale, 2012) and at least 20,000 human proteins (Gaudet et al., 2017; Consortium, 2019), highlighting the urgent need for *in silico* protein-ligand docking methods. Furthermore, a fast and accurate docking tool capable of predicting binding poses for molecules yet to be synthesized would empower mass-scale virtual screening (Lyu et al., 2019), a vital step in modern structure-based drug discovery (Ferreira et al., 2015). It also provides pharmaceutical scientists with an interpretable, information-rich result.

Being a crucial task, predicting the docked pose of a ligand is also a challenging one. Traditional docking methods (Halgren et al., 2004; Morris et al., 1996; Trott & Olson, 2010; Coleman et al., 2013) rely on physics-inspired scoring functions and extensive conformation sampling to obtain the predicted binding pose. Some deep learning methods focus on learning a more accurate scoring function (McNutt et al., 2021; Méndez-Lucio et al., 2021), but at the cost of even lower inference speed due to their adoption of the sampling-scoring framework. Distinct from the above methods, TankBind (Lu et al., 2022) drops the burden of conformation sampling by predicting the protein-ligand distance map, then converting the distance map to a docked pose using gradient descent. The

---

*Equal contribution

optimization objective is the weighted sum of the protein-ligand distance error with respect to the predicted distance map and the intra-ligand distance error w.r.t. the reference intra-ligand distances. This two-stage approach might run into problems during the distance-to-coordinate transformation, as the predicted distance map is, in many cases, not a valid Euclidean distance matrix (Liberti et al., 2014). Recently, Stärk et al. (2022) proposed EquiBind, an equivariant model that directly predicts the coordinates of the docked pose. EquiBind updates the ligand conformation with a graph neural network, then roto-translates the ligand into the pocket using a key-point alignment mechanism. It enjoys significant speedup compared to the popular docking baselines (Hassan et al., 2017; Koes et al., 2013) and provides good pose initializations for them, but on its own the docking performance is less satisfactory. This is probably because, after the one-shot roto-translation, the ligand might fall into an unfavorable position but its conformation could not be further refined.

In this paper, we move one step forward in this important direction and propose E3Bind, the first end-to-end equivariant network that iteratively docks the ligand into the binding pocket. Inspired by AlphaFold2 (Jumper et al., 2021), our model comprises a feature extractor named Trioformer and an iterative coordinate refinement module. The Trioformer encodes the protein and ligand graphs into three information-rich embeddings: the protein residue embeddings, the ligand atom embeddings and the protein-ligand pair embeddings, where the pair embeddings are fused with geometry awareness to enforce the implicit constraints in docking. Our coordinate refinement module decodes the rich representations into E(3)-equivariant coordinate updates. The iterative coordinate update scheme feeds the output pose of a decoder block as the initial pose of the next one, allowing the model to dynamically sense the local context and fix potential errors (see Figure 3). We further propose a self-confidence predictor to select the final pose and evaluate the soundness of our predictions. E3Bind is trained end-to-end with loss directly defined on the output ligand coordinates, relieving the burden of conformation sampling or distance-to-coordinate transformation. Our contributions can be summarized as follows:

- We formulate the docking problem as an iterative refinement process where the model updates the ligand coordinates based on the current context at each iteration.

- We propose an end-to-end E(3) equivariant network to generate the coordinate updates. The network comprises an expressive geometric-aware encoder and an equivariant context-aware coordinate update module.

- Quantitative results show that E3Bind outperforms both traditional score-based methods and recent deep learning models.

## 2 RELATED WORKS

**Protein-ligand docking.** Traditional approaches to protein-ligand docking (Morris et al., 1996; Halgren et al., 2004; Coleman et al., 2013) mainly adopt a sampling, scoring, ranking, and fine-tuning paradigm, with AutoDock Vina (Trott & Olson, 2010) being a popular example. Each part of the docking pipeline has been extensively studied in literature to increase both accuracy and speed (Durrant & McCammon, 2011; Liu et al., 2013; Hassan et al., 2017; Zhang et al., 2020). Multiple subsequent works use deep-learning on 3D voxels (Ragoza et al., 2017; Francoeur et al., 2020; McNutt et al., 2021; Bao et al., 2021) or graphs (Méndez-Lucio et al., 2021) to improve the scoring functions. Nevertheless, these methods are inefficient in general, often taking minutes or even more to predict the docking poses of a single protein-ligand pair, which hinders the accessibility of large-scale virtual screening experiments.

Recently, methods that directly model the distance geometry between protein-ligand pairs have been investigated (Masters et al., 2022; Lu et al., 2022; Zhou et al., 2022). They adopt a two-stage approach for docking, and generate docked poses from predicted protein-ligand distance maps using post-optimization algorithms. Advanced techniques in geometric deep learning, e.g. triangle attention (Jumper et al., 2021) with geometric constraints, have been leveraged to encourage the local geometrical consistency of the distance map (Lu et al., 2022). To bypass the error-prone two-stage framework, EquiBind (Stärk et al., 2022) proposes a fully differentiable equivariant model, which directly predicts coordinates of docked poses with a novel attention-based key-point alignment mechanism (Ganea et al., 2021b). Despite being more efficient, EquiBind fails to beat popular docking baselines on its own, stressing the importance of increasing model expressiveness.

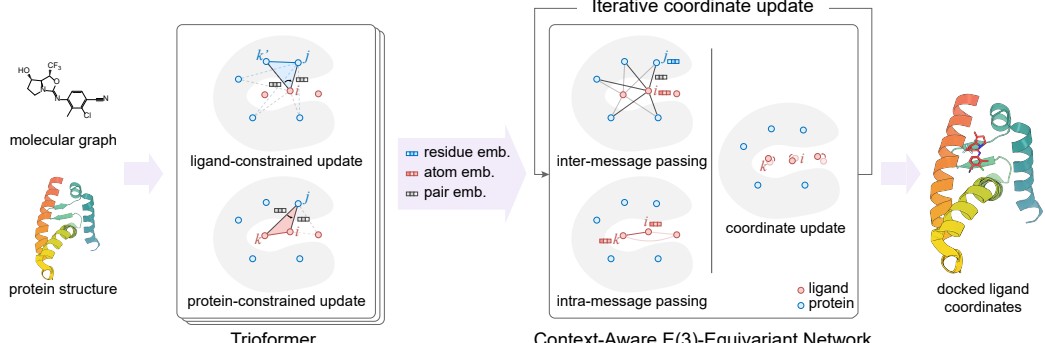

Figure 1: E3Bind model overview. E3Bind takes in a molecular graph of the ligand and a protein structure as input, and extracts features with the Trioformer, which include geometry-aware pair update modules that constrain the embeddings with known intra-protein and intra-ligand distances. E3Bind then iteratively updates the ligand pose with a decoder network to produce the final pose. The context-aware network performs inter- and intra-message passing before generating an E(3)-equivariant coordinate update.

**Molecular conformation generation.** Deep learning has made great progress in predicting low-energy conformations given molecular graphs (Shi et al., 2021; Ganea et al., 2021a; Xu et al., 2022; Jing et al., 2022). State-of-the-art models are generally SE(3)-invariant or equivariant, adopting the score-matching (Vincent, 2011; Song & Ermon, 2019) or diffusion (Ho et al., 2020) framework. Though protein-ligand docking also aims to generate the conformation of a molecule, the bound ligand, applying standard molecular conformation generation approaches is not viable because (1) there are much more atoms in the system and (2) the protein context must be carefully considered when generating the docked pose. Here we model the protein at the residue level and design our model to carefully capture the protein-ligand interactions.

**Protein structure prediction.** Predicting protein folds from sequences has long been a challenging task. Wang et al. (2017); Senior et al. (2020a;b) use deep learning to predict the contact map, the distance map and/or the torsion angles between protein residues, and then convert them to coordinates using optimization-based methods. Recently, AlphaFold2 (Jumper et al., 2021) takes a leap forward by adopting an end-to-end approach with iterative coordinate refinement. It consists of an Evoformer to extract information from Multiple Sequence Alignments (MSAs) and structural templates, and a structure module to iteratively update the coordinates. Though this problem is very different from docking which models heterogeneous entities – a fixed protein structure and a ligand molecular graph, in this paper we show that some ideas can be extended.

## 3  THE E3BIND MODEL

E3Bind tackles the protein-ligand docking task with an encoder for feature extraction and a decoder for coordinate update generation. Specifically, the protein and ligand graphs are first encoded by standard graph encoders. Pair embeddings are constructed between every protein residue - ligand atom pair. A geometry-aware Trioformer is proposed to fully mix the protein, ligand and pair embeddings (Section 3.2). With the rich representations at hand, the iterative coordinate refinement module updates the ligand pose by a series of E(3)-equivariant networks to generate the final docked pose (Section 3.3). The model is trained end-to-end and capable of directly generating the docked pose (Section 3.4). An overview of E3Bind is shown in Figure 1.

### 3.1  PRELIMINARIES

**Notation and Input Feature Construction.** Following Lu et al. (2022), the ligand is treated as an atom-level molecular graph $\mathcal{G}^l$ where the edges denote chemical bonds. The ligand node features $\{\boldsymbol{h}_i^l\}_{1 \leq i \leq n_l}$ are calculated by a TorchDrug (Zhu et al., 2022) implementation of the graph isomorphism network (GIN) (Xu et al., 2018), where $n_l$ is the number of atoms in the ligand. Ligand coordinates are denoted as $\{\boldsymbol{x}_i^l\}_{1 \leq i \leq n_l}$. we represent the protein as a residue-level $K$-nearest neighbor graph $\mathcal{G}^p$. Each protein node $\bar{j} \in \{1, \ldots, n_p\}$ has 3D coordinates $\boldsymbol{x}_j^p$ (which corresponds to the position of the $C_\alpha$ atom of the residue), where $n_p$ is the number of residues in the protein. The

protein node features $\{\boldsymbol{h}_j^{\mathrm{p}}\}_{1 \leq j \leq n_{\mathrm{p}}}$ are calculated with a geometric-vector-perceptron-based graph neural network (GVP-GNN) Jing et al. (2020). For each protein residue-ligand atom pair $(i, j)$, we construct a pair embedding $\boldsymbol{z}_{ij}$ via the outer product module (OPM) which takes in protein and ligand embeddings: $\boldsymbol{z}_{ij} = \mathrm{Linear}\left(\mathrm{vec}\left(\mathrm{Linear}(\boldsymbol{h}_i^{\mathrm{l}}) \bigotimes \mathrm{Linear}(\boldsymbol{h}_j^{\mathrm{p}})\right)\right)$. For notation consistency, throughout this paper we will use $i, k$ to index ligand nodes, and $j, k'$ for protein nodes.

**Problem Definition.** Given a molecular graph of the ligand compound and a fixed protein structure, we aim to predict the binding (docked) pose of the ligand compound $\{\boldsymbol{x}_i^{\mathrm{l}}{}^*\}_{1 \leq i \leq n_{\mathrm{l}}}$. We focus on *blind* docking settings, where the binding pocket is not provided and has to be predicted by the model. In the *blind re-docking* setting, the docked ligand conformation is given, but its position and orientation relative to the protein are unknown. In the *flexible blind self-docking* setting, the docked ligand conformation needs to be inferred besides its position and orientation. The model is thus provided with an unbound ligand structure, which could be obtained by running the ETKDG algorithm (Riniker & Landrum, 2015) with RDKit (Landrum et al., 2013).

**Equivariance.** An important inductive bias in protein ligand docking is E(3) equivariance, *i.e.*, if the input protein coordinates are transformed by some E(3) transformation $g \cdot \{\boldsymbol{x}_j^{\mathrm{p}}\}_{1 \leq j \leq n_{\mathrm{p}}} = \{\boldsymbol{R}\boldsymbol{x}_j^{\mathrm{p}} + \boldsymbol{t}\}_{1 \leq j \leq n_{\mathrm{p}}}, \forall j = 1 \ldots n_{\mathrm{p}}$, the predicted ligand coordinates should also be $g$-transformed: $F(g \cdot \{\boldsymbol{x}_j^{\mathrm{p}}\}_{1 \leq j \leq n_{\mathrm{p}}}) = g \cdot F(\{\boldsymbol{x}_j^{\mathrm{p}}\}_{1 \leq j \leq n_{\mathrm{p}}})$, where $F$ is the coordinate prediction model. To inject such symmetry to our model, we adopt a variant of the equivariant graph neural network (EGNN) as the building block in our coordinate update steps (Satorras et al., 2021).

## 3.2 Extracting Geometry-Consistent Information with Trioformer

The E3Bind encoder extracts information-rich ligand atom embeddings $\boldsymbol{h}_i^{\mathrm{l}}$, protein residue embeddings $\boldsymbol{h}_j^{\mathrm{p}}$ and ligand-protein pair representation $\boldsymbol{z}_{ij}$ that captures the subtle protein-ligand interactions. Note that this is a non-trivial task since implicit geometric constraints must be incorporated in the representations. As shown in the figure on the right, the protein-ligand distance $d_{ij}$ and $d_{ik'}$ can not be predicted independently. They are constrained by the intra-protein distance $d_{jk'}$, as the protein structure is considered to be fixed during docking. In other words, by the triangle inequality, if residues $j$ and $k'$ are close, then ligand atom $i$ cannot be both close to $j$ and far from $k'$. The same thing happens when we consider the given intra-ligand distance $d_{ik}$[1]. While using a simple concatenation of protein and ligand embeddings as the final pair embeddings

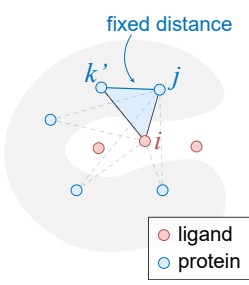

is common among previous methods (Méndez-Lucio et al., 2021; Masters et al., 2022), it dismisses the above geometry constraints and might result in geometrically inconsistent pose predictions.

**Trioformer Overview.** To tackle the above challenge, we propose *Trioformer*, an architecture that intensively updates and mixes the protein, ligand and pair embeddings. In each Trioformer block, we first update the protein and ligand node embeddings with multi-head cross-attention using the pair embeddings as attention bias. Next, we use protein and ligand embeddings to update the pair embeddings via the outer product module: $\boldsymbol{z}_{ij} = \boldsymbol{z}_{ij} + \mathrm{OPM}\left(\boldsymbol{h}_i^{\mathrm{l}}, \boldsymbol{h}_j^{\mathrm{p}}\right)$. The pair embeddings then go through geometry-aware pair update modules described below to produce geometry-consistent representations. The final block outputs are the multi-layer perceptron (MLP)-transitioned protein, ligand and pair embeddings. Details of the Trioformer are described in Section B.

**Geometry-Aware Pair Updates.** To inject geometry awareness to the pair embeddings, we construct intra-ligand and intra-protein distance embeddings ($\boldsymbol{d}_{ik}^*$ and $\boldsymbol{d}_{jk'}^*$ respectively) and then use them to update the pair embeddings. Following (Jumper et al., 2021; Lu et al., 2022), the edge updates are arranged in the form of triangles comprising two ligand-protein edges and one intra-edge (intra-ligand or intra-protein edge) where the distance constraints apply.

---

[1]During docking, the bond lengths, bond angles and conformation of small rings are mostly unchanged, while torsion angles of rotatable bonds might change drastically (Trott & Olson, 2010; Méndez-Lucio et al., 2021; Stärk et al., 2022). In the flexible docking setting, we provide distance $d_{ik}^*$ of atoms $i$ and $k$ in the *unbound* ligand structure predicted by ETKDG (Riniker & Landrum, 2015) to constrain the model if $i, k$ are $\leq$ 2-hop neighbors or members of the same ring. Distance of all other ligand atom pairs are not provided.

For protein-constrained attentive pair updates, each pair $(i, j)$ attends to all neighboring pairs $\{(i, k')\}_{1 \le k' \le n_{\mathrm{p}}}$ with a common ligand node $i$. We compute a geometry-informed attention weight $a_{ijk'}^{(h)}$ for each neighbor $(i, k')$, by adding an attention bias $t_{jk'}^{(h)} = \mathrm{Linear}^{(h)}(\boldsymbol{d}_{jk'})$ computed from the intra-protein distance embeddings per attention head $h$. This geometry-informed attention weight is a key ingredient of Trioformer that differentiates it from the AlphaFold2's Evoformer Jumper et al. (2021). We then perform a standard multi-head attention with $H$ heads to aggregate information from neighboring pairs,

$$a_{ijk'}^{(h)} = \mathrm{softmax}_{k'} \left( \frac{1}{\sqrt{c}} \boldsymbol{q}_{ij}^{(h)\top} \boldsymbol{k}_{ik'}^{(h)} + b_{ij}^{(h)} + t_{jk'}^{(h)} \right), \tag{1}$$

$$\boldsymbol{z}_{ij} = \boldsymbol{z}_{ij} + \mathrm{Linear} \left( \mathrm{concat}_{1 \le h \le H} \left( \boldsymbol{g}_{ij}^{(h)} \odot \sum_{k'=1}^{n_{\mathrm{p}}} a_{ijk'}^{(h)} \boldsymbol{v}_{ik'}^{(h)} \right) \right), \tag{2}$$

where $\boldsymbol{g}_{ij}^{(h)} = \sigma(\mathrm{Linear}^{(h)}(\boldsymbol{z}_{ij}))$ is a per-head output gate, $\boldsymbol{q}_{ij}^{(h)}, \boldsymbol{k}_{ij}^{(h)}, \boldsymbol{v}_{ij}^{(h)}, b_{ij}^{(h)}$ are linear projections of the pair embedding $\boldsymbol{z}_{ij}$, and $t_{jk'}^{(h)}$ is a distance-based bias described above.

The ligand-constrained attentive pair updates are designed similarly. Here, for pair $(i, j)$, the neighboring pairs $\{(k, j)\}_{1 \le k \le n_{\mathrm{l}}}$ are those sharing the same protein node $j$. Multi-head attention is performed across all such neighbors with constraints from intra-ligand distances $\boldsymbol{d}_{ik}$.

## 3.3 Iterative Coordinate Update with Context-Aware E(3)-Equivariant Layer

The coordinate update module iteratively adjusts the current ligand structure $\{\boldsymbol{x}_i^!\}_{1 \le i \le n_{\mathrm{l}}}$ towards the docked pose based on the extracted representations. This module is designed to satisfy the following desiderata: (1) **protein context awareness**: the model must have an in-depth understanding of the interaction between the ligand and its protein context in order to produce a protein-ligand complex with optimized stability (*i.e.* lowest energy); (2) **self (ligand) context awareness**: the model should honor the basic geometric constraints of the ligand so that the predicted ligand conformation is physically valid; (3) **E(3)-equivariance**: if both the protein and the input ligand pose are transformed by an E(3) transformation, the ligand should dock into the same pocket with the same pose, with its coordinates transformed with the same E(3) transformation. Compared with methods that generate the final pose in one shot, iterative refinement allows the model to dynamically sense the local context and correct potential errors as the ligand gradually moves toward its final position.

We start from an unbound ligand structure[2]. At iteration step $t = 0, \ldots, T-1$, the module updates the protein, ligand and pair representations (summarized by $\boldsymbol{h}^{(t)}$) and the ligand coordinates $\{\boldsymbol{x}_i^{(t)}\}_{1 \le i \le n_{\mathrm{l}}}$ with a context-aware E(3)-equivariant layer,

$$\left( \boldsymbol{h}^{(t+1)}, \{\Delta\boldsymbol{x}_i^{(t+1)}\}_{1 \le i \le n_{\mathrm{l}}} \right) = \mathrm{DecoderLayer}^{(t)} \left( \boldsymbol{h}^{(t)}, \{\boldsymbol{x}_i^{(t)}\}_{1 \le i \le n_{\mathrm{l}}}, \{\boldsymbol{x}_j\}_{1 \le j \le n_{\mathrm{p}}} \right), \tag{3}$$

$$\boldsymbol{x}_i^{(t+1)} = \boldsymbol{x}_i^{(t)} + \Delta\boldsymbol{x}_i^{(t+1)}, \qquad 0 \le t < T, \tag{4}$$

where $\{\boldsymbol{x}_j\}_{1 \le j \le n_{\mathrm{p}}}$ are the fixed protein coordinates. We now introduce $\mathrm{DecoderLayer}^{(t)}$ in detail.

**Context-Aware Message Passing.** To ensure the coordinate update module captures both the protein and ligand context, we construct a heterogeneous context graph comprising both protein and ligand nodes. In the graph, *inter-edges* connect protein and ligand nodes, while *intra-edges* connect two ligand nodes (Figure 1). We perform inter- and intra-edge message passing on the graph to explore the current context:

$$\boldsymbol{h}_i^{(t+1)} = \boldsymbol{h}_i^{(t)} + \sum_{j=1}^{n_{\mathrm{p}}} \boldsymbol{m}_{ij}^{(t)} + \sum_{k=1}^{n_{\mathrm{l}}} \boldsymbol{m}_{ik}^{(t)}, \tag{5}$$

$$\boldsymbol{h}_j^{(t+1)} = \boldsymbol{h}_j^{(t)} + \sum_{i=1}^{n_{\mathrm{l}}} \boldsymbol{m}_{ji}^{(t)}. \tag{6}$$

---

[2]In the rigid docking setting, this structure has the same conformation as the docked ligand structure.

Note that the three types of messages are generated using different sets of parameters.

**Equivariant Coordinate Update.** We use the *equivariant graph convolution layer* (EGCL) to process current context geometry and update ligand coordinates in an E(3)-equivariant manner. Specifically, we compute messages from the node (and edge) representations and distance information using MLPs $\phi^m$ and $\varphi^m$,

$$\left(\boldsymbol{m}_{ij}^{(t)}, \boldsymbol{m}_{ji}^{(t)}\right) = \phi^m\left(\boldsymbol{z}_{ij}, \boldsymbol{h}_i^{(t)}, \boldsymbol{h}_j^{(t)} \left\|\boldsymbol{x}_i^{(t)} - \boldsymbol{x}_j^{(t)}\right\|\right), \tag{7}$$

$$\boldsymbol{m}_{ik}^{(t)} = \varphi^m\left(\boldsymbol{h}_i^{(t)}, \boldsymbol{h}_k^{(t)} \left\|\boldsymbol{x}_i^{(t)} - \boldsymbol{x}_k^{(t)}\right\|\right). \tag{8}$$

We generate equivariant coordinate updates using the following equation,

$$\Delta\boldsymbol{x}_i^{(t)} = \sum_{j=1}^{n_{\mathrm{p}}} \frac{\boldsymbol{x}_j^{(t)} - \boldsymbol{x}_i^{(t)}}{\|\boldsymbol{x}_j^{(t)} - \boldsymbol{x}_i^{(t)}\|} \phi^x(\boldsymbol{m}_{ij}^{(t)}) + \sum_{k=1}^{n_{\mathrm{l}}} \frac{\boldsymbol{x}_k^{(t)} - \boldsymbol{x}_i^{(t)}}{\|\boldsymbol{x}_k^{(t)} - \boldsymbol{x}_i^{(t)}\|} \varphi^x(\boldsymbol{m}_{ik}^{(t)}), \tag{9}$$

where $\phi^x$ and $\varphi^x$ are gated MLPs that evaluate the message importance. Ligand and protein node embeddings are updated using equations 5 and 6, respectively.

Note that our decoder is also compatible with other equivariant graph layers, such as the geometric vector perceptron (GVP) (Jing et al., 2020) or vector neuron (Deng et al., 2021). We choose EGCL as it is a powerful layer for molecular modeling and conformation generation (Satorras et al., 2021; Huang et al., 2022; Hoogeboom et al., 2022).

**Self-Confidence Prediction.** At the end of our decoder, an additional self-confidence module is added to predict the model's confidence for its predicted docked pose (Jumper et al., 2021). The confidence is a value from zero to one calculated by the equation $\hat{c} = \sigma\left(\mathrm{MLP}\left(\sum_{i=1}^{n_{\mathrm{l}}} \boldsymbol{h}_i^{(T)}\right)\right)$.

### 3.4 TRAINING AND INFERENCE

**End-to-End Training.** E3Bind directly generates the predicted ligand coordinates along with a confidence score. We define a simple coordinate loss and train the model end-to-end: $\mathcal{L}_{\mathrm{coord}} = \sum_{i=1}^{n_{\mathrm{l}}} \left(\boldsymbol{x}_i - \boldsymbol{x}_i^*\right)^2$ , where $\boldsymbol{x}_i$ is the predicted coordinates and $\boldsymbol{x}_i^*$ is the ground truth. We train the self-confidence module with $\mathcal{L}_{\mathrm{confidence}} = \mathrm{MSE}\left(\hat{c}, c^*\right)$, where MSE is the mean squared error loss and $c^*$ is the self-confidence prediction target, *i.e.* the (detached) root-mean-square deviation (RMSD) of the predicted coordinates. Details are deferred to Section C.1. Intuitively, we want our model to predict a low $\hat{c}$ for high-RMSD predictions. The final training loss is the combination of the coordinate loss and the confidence loss $\mathcal{L} = \mathcal{L}_{\mathrm{coord}} + \beta\mathcal{L}_{\mathrm{confidence}}$, where $\beta$ is a hyperparameter. This loss aligns with the goal of docked pose prediction and avoids the time-consuming sampling process and potentially error-prone distance-to-coordinate transformation.

**Inference Process.** In practice, the target protein may contain multiple binding sites, or be extremely large where most parts are irrelevant for ligand binding. To solve this problem, we use P2Rank (Krivák & Hoksza, 2018) to segment protein to less than 10 functional blocks (defined as a 20Å graph around the block center), following TankBind (Lu et al., 2022). We then initialize the unbound/bound (depending on the setting) ligand structure with random rotation and translation in each block, dock the ligand, and select the predicted docked pose with the highest self-confidence. Note that the initial pose might clash with the protein, but these clashes are likely to be repaired during the iterative refinement process. Different from TankBind which selects the final pose through binding affinity estimation for all functional block's predictions, our pose selection is based on self-confidence. As a result, our model only feeds on protein-ligand complex structures, leaving the potential of including more structures without paired affinity for training.

## 4 EXPERIMENTS

### 4.1 FLEXIBLE SELF DOCKING

As E3Bind is designed to model the flexibility of ligand conformation, it is natural to evaluate it in the flexible blind self-docking setting. We defer the blind re-docking results to Appendix Section A.

| | LIGAND RMSD | | | | % Below ↑ | | CENTROID DISTANCE | | | | % Below ↑ | |
| | Percentiles ↓ | | | | | | Percentiles ↓ | | | | | |
| Method | 25% | 50% | 75% | Mean | 2Å | 5Å | 25% | 50% | 75% | Mean | 2Å | 5Å |
|---|---|---|---|---|---|---|---|---|---|---|---|---|
| QVina-W | 2.5 | 7.7 | 23.7 | 13.6 | 20.9 | 40.2 | 0.9 | 3.7 | 22.9 | 11.9 | 41.0 | 54.6 |
| GNINA | 2.8 | 8.7 | 22.1 | 13.3 | 21.2 | 37.1 | 1.0 | 4.5 | 21.2 | 11.5 | 36.0 | 52.0 |
| SMINA | 3.8 | 8.1 | 17.9 | 12.1 | 13.5 | 33.9 | 1.3 | 3.7 | 16.2 | 9.8 | 38.0 | 55.9 |
| GLIDE | 2.6 | 9.3 | 28.1 | 16.2 | 21.8 | 33.6 | 0.8 | 5.6 | 26.9 | 14.4 | 36.1 | 48.7 |
| Vina | 5.7 | 10.7 | 21.4 | 14.7 | 5.5 | 21.2 | 1.9 | 6.2 | 20.1 | 12.1 | 26.5 | 47.1 |
| EquiBind | 3.8 | 6.2 | 10.3 | 8.2 | 5.5 | 39.1 | 1.3 | 2.6 | 7.4 | 5.6 | 40.0 | 67.5 |
| TankBind | 2.6 | 4.2 | **7.6** | 7.8 | 17.6 | 57.8 | **0.8** | 1.7 | 4.3 | 5.9 | 55.0 | 77.8 |
| E3Bind | **2.1** | **3.8** | 7.8 | **7.2** | **23.4** | **60.0** | **0.8** | **1.5** | **4.0** | **5.1** | **60.0** | **78.8** |
| EquiBind-U | 3.3 | 5.7 | 9.7 | 7.8 | 7.2 | 42.4 | 1.3 | 2.6 | 7.4 | 5.6 | 40.0 | 67.5 |
| TankBind-U | 3.9 | 7.7 | 13.6 | 10.5 | 8.0 | 34.7 | 1.3 | 3.0 | 8.2 | 6.6 | 40.5 | 66.4 |
| E3Bind-U | **2.0** | **3.8** | **7.7** | **7.2** | **25.6** | **60.6** | **0.8** | **1.5** | **4.0** | **5.1** | **59.0** | **78.8** |

Table 1: Flexible blind self-docking performance. Models with "-U" suffix do not perform post-optimization steps that further enforce intra-ligand geometry constraints.

**Data.** We use the PDBbind v2020 dataset (Liu et al., 2017) for training and evaluation. We follow the time dataset split from (Stärk et al., 2022), where 363 complex structures uploaded later than 2019 serve as test examples. After removing structures sharing ligands with the test set, the remaining 16739 structures are used for training and 968 structures are used for validation.

**Baselines.** We compare E3Bind with recent deep learning (DL) models and a set of traditional score-based methods. For recent deep learning models, TankBind (Lu et al., 2022) and EquiBind (Stärk et al., 2022) are included. For score-based methods, QVina-W (Hassan et al., 2017), GNINA (Mc-Nutt et al., 2021), SMINA (Koes et al., 2013) and GLIDE (Halgren et al., 2004) are included.

We additionally distinguish between the uncorrected and corrected versions of the recent deep learning models following EquiBind. The corrected versions adopt post optimization methods (e.g. gradient descent in TankBind (Lu et al., 2022), fast point cloud fitting in EquiBind (Stärk et al., 2022)) to further enforce the intra-ligand geometry constraints when generating the predicted structure. The uncorrected versions (suffixed with -U), which do not perform post-optimization, reflect the model's own capability for pose prediction. Specifically, EquiBind-U and E3Bind-U outputs are directly generated by the neural networks and TankBind-U outputs are optimized from the predicted distance map without the ligand configuration loss. For a fair comparison, E3Bind results are refined from the E3Bind-U predicted coordinates by the post-optimization method of TankBind.

**Metrics.** We evaluate the quality of the generated ligand pose by the following metrics: (1) **Ligand RMSD**, the root-mean-square deviation of the ligand's Cartesian coordinates, measures at the atom level how well the model captures the protein-ligand binding mechanisms; (2) **Centroid Distance**, defined as the distance between the average coordinates of predicted and ground-truth ligand structures, reflects the model's capacity to find the binding site. All metrics are calculated with hydrogen atoms discarded following previous work.

**Performance in Flexible Self-Docking.** Table 1 summarized the quantitative results in flexible blind self-docking. Our model achieves state-of-the-art on most metrics. Specifically, E3Bind shows exceptional power in finding ligand poses with high resolution, where the percentage of predicted poses with $\text{RMSD} < 2\,\text{Å}$ increases by 33% compared to the previous state-of-the-art DL-based model TankBind. E3Bind achieves 2.1 Å for the 25-th percentile of the ligand RMSD, ourperforming all previous methods by a large margin. These results verify that our model is able to better capture the protein-ligand interactions and generate a geometrically-consistent binding pose. Notably, among uncorrected deep learning models, E3Bind-U enjoys more significant performance improvement, showcasing its low dependency on additional post-optimization. E3Bind-U also outperforms traditional docking softwares by orders of magnitude in inference speed (Section G), demonstrating its potential for high-throughput virtual screening.

**Performance in Flexible Self Docking for Unseen Protein.** We further evaluate our model's capacity on a subset of the above test set containing 144 complexes with the protein unseen in training. As shown in Table 2, E3Bind and E3Bind-U show better generalization ability than other DL-based model. Note that in this setting traditional score-based methods performs better than DL-based

| | LIGAND RMSD | | | | | | CENTROID DISTANCE | | | | | |
| | Percentiles ↓ | | | | % Below ↑ | | Percentiles ↓ | | | | % Below ↑ | |
| **Method** | 25% | 50% | 75% | Mean | 2 Å | 5 Å | 25% | 50% | 75% | Mean | 2 Å | 5 Å |
|---|---|---|---|---|---|---|---|---|---|---|---|---|
| QVina-W | 3.4 | 10.3 | 28.1 | 16.9 | 15.3 | 31.9 | 1.3 | 6.5 | 26.8 | 15.2 | 35.4 | 47.9 |
| GNINA | 4.5 | 13.4 | 27.8 | 16.7 | 13.9 | 27.8 | 2.0 | 10.1 | 27.0 | 15.1 | 25.7 | 39.5 |
| SMINA | 4.8 | 10.9 | 26.0 | 15.7 | 9.0 | 25.7 | 1.6 | 6.5 | 25.7 | 13.6 | 29.9 | 41.7 |
| GLIDE | 3.4 | 18.0 | 31.4 | 19.6 | **19.6** | 28.7 | **1.1** | 17.6 | 29.1 | 18.1 | 29.4 | 40.6 |
| Vina | 7.9 | 16.6 | 27.1 | 18.7 | 1.4 | 12.0 | 2.4 | 15.7 | 26.2 | 16.1 | 20.4 | 37.3 |
| EquiBind | 5.9 | 9.1 | 14.3 | 11.3 | 0.7 | 18.8 | 2.6 | 6.3 | 12.9 | 8.9 | 16.7 | 43.8 |
| TankBind | 3.4 | **5.7** | 10.8 | 10.5 | 3.5 | **43.7** | 1.2 | 2.6 | 8.4 | 8.2 | 40.9 | **70.8** |
| E3Bind | **3.0** | 6.1 | **10.2** | 10.1 | 6.3 | 38.9 | 1.2 | **2.3** | **7.0** | 7.6 | 43.8 | 66.0 |
| EquiBind-U | 5.7 | 8.8 | 14.1 | 11.0 | 1.4 | 21.5 | 2.6 | 6.3 | 12.9 | 8.9 | 16.7 | 43.8 |
| TankBind-U | 4.0 | 7.9 | 14.9 | 8.3 | 3.5 | 34.0 | 1.4 | 3.3 | 10.9 | 8.3 | 35.4 | 65.2 |
| E3Bind-U | **3.1** | **6.0** | **10.6** | 10.1 | **5.6** | 41.0 | **1.2** | **2.3** | **7.8** | 7.7 | **42.4** | **65.3** |

Table 2: Flexible blind self-docking performance on unseen receptors. Test set contains 144 complexes with the protein not seen in training.

models on finding high-quality binding poses ($\text{RMSD} < 2\,\text{Å}$), highlighting the need of increasing the out-of-distribution generalization capability for the latter. That said, we observe that E3Bind produces much fewer predictions that are far off-target ($\text{RMSD} > 5\,\text{Å}$).

**Benefit of Iterative Refinement.** Figure 2 demonstrates the benefits of iterative refinement in improving ligand pose prediction. As the ligand undergoes more rounds of coordinate refinement, its RMSD decreases and self-confidence score increases. Figure 3 visualizes how E3Bind's iterative refinement process successfully identified the correct binding site after 4 iterations and further refined the ligand conformation for maximal interactions with the protein. The final pose is close to the ground truth with an RMSD of 1.42 Å. In contrast, Equibind, a method that does not adopt the iterative refinement scheme, yields a pose with an RMSD of 4.18 Å.

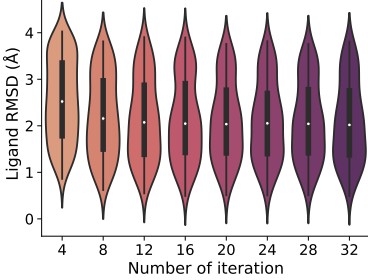 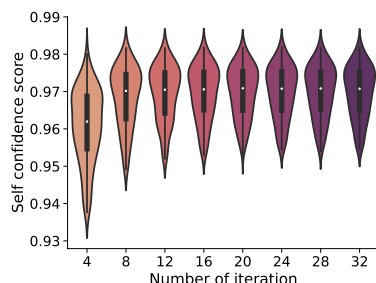

Figure 2: Violin plots of the ligand RMSD (in Å, left) and the self-confidence score (right) versus the number of coordinate update iterations (top 50% examples shown).

## 4.2 ABLATION STUDY

A series of ablation studies are done to investigate different factors influencing the performance (Section E.1). (1) Removing geometry constraints in the pair update modules degrades the fraction of good poses (with $\text{RMSD} < 2\,\text{Å}$) from 23.4% to 20.1%. (2) Replacing the Trioformer with a simple concatenation of the protein and ligand embeddings hurts the performance even worse (11.7%), highlighting the strength of Trioformer in extracting geometric features for docking. (3) Decreasing the number of iterations from 32 to 4 makes the fraction of good poses drop to 14.9%, reiterating the benefit of the iterative refinement process. (4) The performance degrades to 17.6% when we remove intra-message passing in the coordinate update modules. (5) Removing P2Rank and instead segmenting the protein into 30 random blocks for docking has little impact on performance, showing that E3Bind does not rely on P2Rank as much as TankBind for binding site pre-selection.

## 4.3 CASE STUDY

**E3Bind correctly identifies the binding site in an unseen large protein.** Figure 4a shows a representative case of a challenging example in blind docking. For an unseen protein, E3Bind (shown in magenta) correctly identifies the native binding site with a low RMSD of 4.2 Å, while QVina-W,

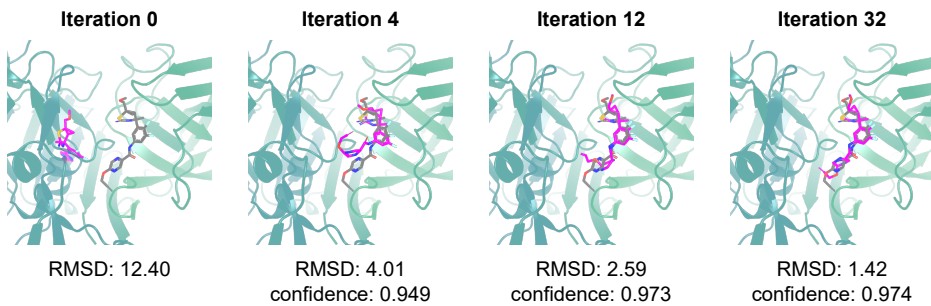

| Iteration 0 | Iteration 4 | Iteration 12 | Iteration 32 |
|---|---|---|---|
| RMSD: 12.40 | RMSD: 4.01
confidence: 0.949 | RMSD: 2.59
confidence: 0.973 | RMSD: 1.42
confidence: 0.974 |

Figure 3: E3Bind coordinate refinement trajectory for ligand in PDB 6PZ4. In each figure, the ground-truth docked ligand pose is shown in gray and the initialized (Iteration 0) / predicted (Iterations 4, 12, 32) structures in magenta. RMSD and model confidence are written below the figures.

EquiBind and TankBind dock to three distinct binding sites away from the ground truth. Further examination of the self-confidence score shows this pose is the only one with confidence above 0.91, while poses at other binding sites all have confidence below 0.86 (Figure S2).

**E3Bind strikes a balance between modeling protein-ligand interaction and structure validity.** Figure 4b presents the docking result of a small protein. All methods except TankBind identify the correct binding site, with E3Bind prediction aligning better to the ground truth protein-ligand interaction pattern. Interestingly, TankBind produces a knotted structure that is invalid in nature. This shows that the two-stage approach might generate invalid protein-ligand distance maps that can not be transformed into plausible structures in the distance-to-coordinate optimization stage.

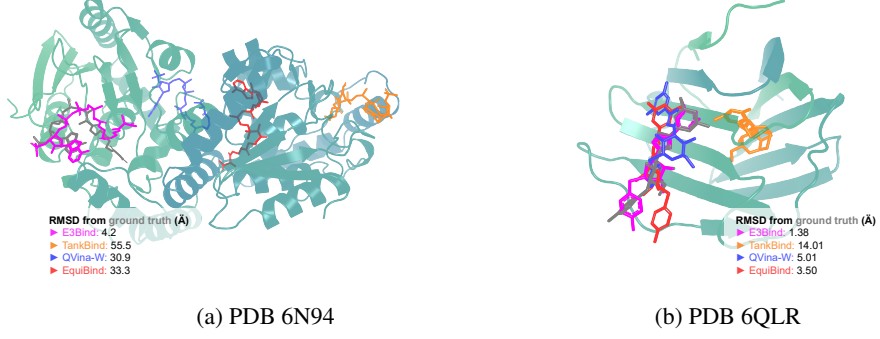

(a) PDB 6N94

(b) PDB 6QLR

Figure 4: Case studies. Predicted pose from E3Bind (magenta), TankBind (orange), EquiBind (red) and QVina-W (blue) are placed together with the target protein. RMSDs from ground truth ligand pose (grey) are shown in the figure. (a) E3Bind correctly identifies the binding site from the large protein, while the other methods are off-site. (b) E3Bind outperformed the other models in predicting the binding pose accurately, while TankBind generated an invalid structure with knotted rings.

## 5 CONCLUSION

Fast and accurate docking methods are vital tools for small molecule drug research and discovery. This work proposes E3Bind, an end-to-end equivariant network for protein-ligand docking. E3Bind predicts the docked ligand pose through a feature extraction – coordinate refinement pipeline. Geometry-consistent protein, ligand and pair representations are first extracted by Trioformer. Then the ligand coordinates are iteratively updated by a context-aware E(3)-equivariant network. Empirical experiments show that E3Bind is competitive against state-of-the-art blind docking methods, especially without post-optimization of the pose. Interesting future directions include: modeling the protein backbone/side-chain dynamics to better capture the drug-target interaction and exploring better ways of feature extraction, geometry constraint incorporation and coordinate refinement.

ACKNOWLEDGEMENT

We would like to express sincere thanks to Bozitao Zhong. He contributed a lot to this project. We would also like to extend our appreciation to Minkai Xu, Chang Ma, Haoxiang Yang, Minghao Xu, Zhaocheng Zhu for their helpful discussions and insightful comments. This project is supported by Twitter, Intel, the Natural Sciences and Engineering Research Council (NSERC) Discovery Grant, the Canada CIFAR AI Chair Program, Samsung Electronics Co., Ltd., Amazon Faculty Research Award, Tencent AI Lab Rhino-Bird Gift Fund, a NRC Collaborative R&D Project (AI4D-CORE-06) as well as the IVADO Fundamental Research Project grant PRF-2019-3583139727.

REPRODUCIBILITY STATEMENT

All code for data preprocessing, training and inference will be publicly released upon acceptance. Trioformer detail can be found in Section B. Detail of the confidence loss is in Section C.1. Other implementation details are in Section D.

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

## A    EXPERIMENT FOR BLIND RE-DOCKING

The setting of blind re-docking allows the model to take the ground-truth conformation as prior knowledge. Note that this is a less realistic setting, since in many real-world applications the bound ligand conformation is not known in advance. Compared to the flexible blind self-docking, the model is given the full ligand conformation as geometric constraints, and only needs to predict the rigid-body motion for the molecule to dock into the binding site. Although the nature of rigid docking is somewhat contradictory to the E3Bind model as it allows the full flexibility of the ligand structure, we are still able to enforce a rigid structure through TankBind-style gradient descent post-optimization (Lu et al., 2022) using the ground truth conformation as configuration loss.

| | LIGAND RMSD | | | | | | CENTROID DISTANCE | | | | | |
| | Percentiles ↓ | | | | % Below ↑ | | Percentiles ↓ | | | | % Below ↑ | |
| **Method** | 25% | 50% | 75% | Mean | 2 Å | 5 Å | 25% | 50% | 75% | Mean | 2 Å | 5 Å |
|---|---|---|---|---|---|---|---|---|---|---|---|---|
| QVina-W | 1.6 | 7.9 | 24.1 | 13.4 | 27.7 | 39.0 | 0.9 | 3.8 | 23.2 | 11.8 | 40.4 | 55.4 |
| GNINA | 1.3 | 6.1 | 22.9 | 12.2 | 32.2 | 46.8 | 0.7 | 2.8 | 22.1 | 10.9 | 43.8 | 58.4 |
| SMINA | 1.4 | 6.2 | 15.2 | 10.3 | 30.1 | 46.7 | 0.8 | 2.6 | 12.7 | 8.5 | 45.3 | 63.5 |
| GLIDE | **0.5** | 8.3 | 29.5 | 15.7 | **43.4** | 45.7 | **0.3** | 4.9 | 28.5 | 14.8 | 45.4 | 50.4 |
| Vina | 4.5 | 9.7 | 19.9 | 13.4 | 13.2 | 26.7 | 1.7 | 5.5 | 18.7 | 11.2 | 29.8 | 47.9 |
| EquiBind-R | 2.0 | 5.1 | 9.8 | 7.4 | 25.1 | 49.0 | 1.4 | 2.6 | 7.3 | 5.8 | 40.8 | 66.9 |
| TankBind | 1.4 | 3.4 | **7.0** | 7.0 | 37.2 | **63.9** | 0.8 | 1.7 | 4.1 | 5.6 | 55.1 | 78.2 |
| E3Bind | 1.2 | **3.3** | 7.2 | **6.7** | 38.3 | **63.9** | 0.6 | **1.3** | **3.8** | **5.1** | **61.4** | **79.0** |
| TankBind-U | 4.5 | 8.0 | 15.5 | 10.9 | 8.0 | 27.5 | 1.5 | 3.3 | 8.3 | 6.7 | 33.3 | 60.3 |
| E3Bind-U | **1.4** | **3.2** | **7.1** | **6.8** | **34.7** | **65.6** | **0.6** | **1.4** | **3.7** | **5.1** | **61.4** | **79.3** |

Table S1: Blind re-docking performance. EquiBind-R is a variant of EquiBind that does not update the ligand conformation with IEGMN and only predicts a translation and rotation.

Results of blind rigid docking are shown in Table S1. Although the setting of rigid docking is not very suitable for our E3Bind model, we achieve SOTA in most metrics, especially in centroid distance.

## B    DETAILS OF THE TRIOFORMER

The Trioformer comprises a stack of Trioformer blocks detailed in Algorithm 1. It is a variant of AlphaFold2's Evoformer with baked-in geometry awareness. In each block, the ligand embeddings $\{\boldsymbol{h}_i^l\}$ are updated with a multi-head cross attention module, with the transformed pair embeddings as bias (MHAWithPairBias):

$$a_{ik'}^{(h)} = \mathrm{softmax}_k \left( \frac{1}{\sqrt{c}} \boldsymbol{q}_i^{(h)\top} \boldsymbol{k}_{k'}^{(h)} + b_{ik'}^{(h)} \right) \tag{10}$$

$$\boldsymbol{h}_i^l = \boldsymbol{h}_i^l + \mathrm{Linear}\left( \mathrm{concat}_{1 \leq h \leq H} \left( \sum_{k'=1}^{n_\mathrm{p}} a_{ik'}^{(h)} \boldsymbol{v}_{k'} \right) \right) \tag{11}$$

The protein embeddings $\{\boldsymbol{h}_j^\mathrm{p}\}$ are updated likewise. The node-level embeddings then go through a two-layer MLP transition modules and reach the block output state. On another track, the pair embeddings $\{\boldsymbol{z}_{ij}\}$ are first updated with the node-level embeddings via the outer product module. Then, geometry-aware attentive pair update modules (detailed in Section 3.2) injects implicit geometric constraints into the pair embeddings. Finally, the pair embeddings go through MLP transition modules and become the remaining part of block outputs.

## C    DETAILS OF SELF-CONFIDENCE PREDICTION

### C.1    CONFIDENCE LOSS

We train the self-confidence module with a confidence prediction loss $\mathcal{L}_\mathrm{confidence}$, which is the mean squared error (MSE) between the predicted self-confidence and its target value $c^*$. $c^*$ is dependent

---

**Algorithm 1:** Trioformer Block

---

**Input**: ligand embeddings $\{\boldsymbol{h}_i^{\mathrm{l}}\}$, protein embeddings $\{\boldsymbol{h}_j^{\mathrm{p}}\}$, pair embeddings $\{\boldsymbol{z}_{ij}\}$,
  intra-ligand distances $\{d_{ik}\}$, intra-protein distances $\{d_{jk'}\}$

```
/* Cross attention updates ligand and protein with pair */
```
$\{\boldsymbol{h}_i^{\mathrm{l}}\} \leftarrow \{\boldsymbol{h}_i^{\mathrm{l}}\} + \mathrm{MHAWithPairBias}(\{\boldsymbol{h}_i^{\mathrm{l}}\}, \{\boldsymbol{h}_j^{\mathrm{p}}\}, \{\boldsymbol{h}_j^{\mathrm{p}}\}, \{\boldsymbol{z}_{ij}\})$ ;
$\{\boldsymbol{h}_j^{\mathrm{p}}\} \leftarrow \{\boldsymbol{h}_j^{\mathrm{p}}\} + \mathrm{MHAWithPairBias}(\{\boldsymbol{h}_j^{\mathrm{p}}\}, \{\boldsymbol{h}_i^{\mathrm{l}}\}, \{\boldsymbol{h}_i^{\mathrm{l}}\}, \{\boldsymbol{z}_{ij}\})$ ;
```
/* Node level transitions */
```
$\{\boldsymbol{h}_i^{\mathrm{l}}\} \leftarrow \{\boldsymbol{h}_i^{\mathrm{l}}\} + \mathrm{MLP}(\{\boldsymbol{h}_i^{\mathrm{l}}\})$;
$\{\boldsymbol{h}_j^{\mathrm{p}}\} \leftarrow \{\boldsymbol{h}_j^{\mathrm{p}}\} + \mathrm{MLP}(\{\boldsymbol{h}_j^{\mathrm{p}}\})$;
```
/* OPM updates pair embeddings with ligand and protein */
```
$\{\boldsymbol{z}_{ij}\} \leftarrow \{\boldsymbol{z}_{ij}\} + \mathrm{OPM}(\{\boldsymbol{h}_i^{\mathrm{l}}\}, \{\boldsymbol{h}_j^{\mathrm{p}}\})$ ;
```
/* Geometry-aware pair update modules */
```
$\{\boldsymbol{z}_{ij}\} \leftarrow \{\boldsymbol{z}_{ij}\} + \mathrm{LigandConstrainedAttentivePairUpdate}(\{\boldsymbol{z}_{ij}\}, \{d_{ik}\}, \{d_{jk'}\})$ ;
$\{\boldsymbol{z}_{ij}\} \leftarrow \{\boldsymbol{z}_{ij}\} + \mathrm{ProteinConstrainedAttentivePairUpdate}(\{\boldsymbol{z}_{ij}\}, \{d_{ik}\}, \{d_{jk'}\})$ ;
```
/* Pair transition */
```
$\{\boldsymbol{z}_{ij}\} \leftarrow \{\boldsymbol{z}_{ij}\} + \mathrm{MLP}(\{\boldsymbol{z}_{ij}\})$ ;
**return** $\{\boldsymbol{h}_i^{\mathrm{l}}\}, \{\boldsymbol{h}_j^{\mathrm{p}}\}, \{\boldsymbol{z}_{ij}\}$

---

on the root-mean-square deviation (RMSD) of the coordinate prediction: it linearly increases with the RMSD of the predicted pose when the latter is small, and is set to a small value $c_0$ once RMSD reaches $\gamma$:

$$c\left(\{\boldsymbol{x}_i^{\mathrm{l}}\}\right) = \begin{cases} 1 - \frac{1}{2\gamma} \cdot \mathrm{RMSD}\left(\{\boldsymbol{x}_i^{\mathrm{l}}\}, \{\boldsymbol{x}_i^{\mathrm{l}*}\}\right) & \text{if } \mathrm{RMSD}\left(\{\boldsymbol{x}_i^{\mathrm{l}}\}, \{\boldsymbol{x}_i^{\mathrm{l}*}\}\right) \leq \gamma, \\ c_0 & \text{otherwise.} \end{cases} \tag{12}$$

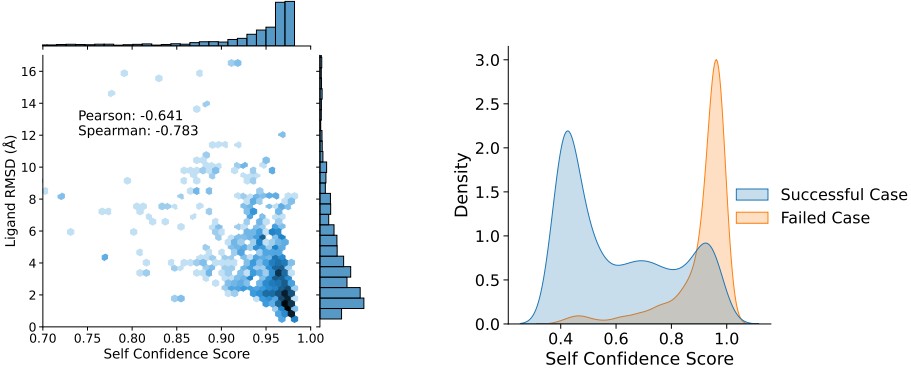

Figure S1: Behavior of the self-confidence model. (Left) Jointplot showing the relationship between the predicted confidence score and ligand RMSD (in Å) for test examples. (Right) E3Bind self-confidence module can distinguish successfully-docked ($\mathrm{RMSD} < 5\text{Å}$) poses from failed ones.

### C.2 SELF-CONFIDENCE PREDICTION RESULTS

We plot the relationship between the predicted confidence score and the ligand RMSD for test examples on the left plot of Figure S1. Most data points have high confidence scores and low ligand RMSDs. We further calculate the correlation metrics between the negative RMSD and the predicted confidence score. The results give a Pearson correlation coefficient of 0.641 and a Spearman correlation coefficient of 0.783, indicating a decent pose-selection capability of our self-confidence prediction module.

We next investigate the behavior of the self-confidence score for success and failure cases (Figure S1, right). The distribution plot shows that the distributions for these two cases have high divergence.

For successful docking examples ($\text{RMSD} < 5\,\text{Å}$), the average confidence score is 0.91, while for failure cases, the average confidence score is 0.61.

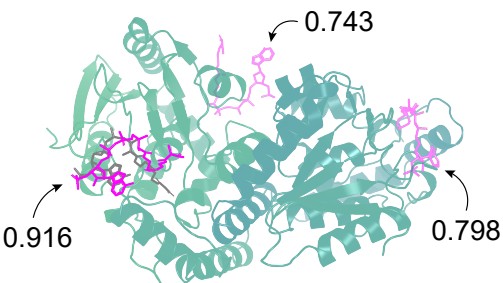

Figure S2: Case study on E3Bind's self-confidence module: PDB 6N94. The ligand pose (in magenta) closest to the ground truth (in grey) has the highest confidence score of 0.916. The confidence scores of the two wrongly predicted ligand poses (in magenta, half-transparent) are both below 0.8.

## D    IMPLEMENTATION DETAILS

The dimensions of protein, ligand and pair embeddings are set to 128. Layer normalization and dropout are applied in each block. For the implementation of EGCL, we use the normalized coordinate differences instead of the original coordinate differences. Specifically, all coordinates are divided by 5 before fed into EGCL and the final coordinates are multiplied by 5. SiLU activation function is used in the EGCL layers and LeakyReLU is used in the Trioformer blocks. All models including variants in ablation study are trained with Adam optimizer with learning rate 0.0001 for 300 epochs. The model with the best valid score (measured by the fraction of predicted pose with $\text{RMSD} < 2\text{Å}$) is evaluated on the test set. We use the TankBind (Lu et al., 2022) pipeline for data preprocessing, protein segmentation and featurization. The only modification is that we remove all complexes in the training set sharing ligands with the test set to reduce information leak at test time.

For the context-aware decoder, instead of using a set of parameters for each EGCL layer, we adopt the recycling technique of AlphaFold2 (Jumper et al., 2021). Specifically, E3Bind's decoder contains 4 EGCL layers (*i.e.* holds parameters for 4 coordinate update iterations). For iterations longer than 4, we recycle the decoder parameters by feeding the previously predicted pose to the decoder as the initial pose. During training, we randomly select the number of cycles from $\text{Uniform}(1, N_{\text{cycle}})$, and only backpropagate through the last cycle (by detaching the gradient of its initial pose) to reduce the computational cost. $N_{\text{cycle}}$ is set to 32 in our experiments.

# E  FURTHER ABLATION STUDY RESULTS

## E.1  ABLATION STUDY OF E3BIND

| | LIGAND RMSD | | | | | | CENTROID DISTANCE | | | | | |
| | Percentiles ↓ | | | | % Below ↑ | | Percentiles ↓ | | | | % Below ↑ | |
| Method | 25% | 50% | 75% | Mean | 2 Å | 5 Å | 25% | 50% | 75% | Mean | 2 Å | 5 Å |
|---|---|---|---|---|---|---|---|---|---|---|---|---|
| E3Bind | 2.1 | 3.8 | 7.8 | 7.2 | 23.4 | 60.0 | 0.8 | 1.5 | 4.0 | 5.1 | 60.0 | 78.8 |
| w/o Geometric Constraint Aware | 2.3 | 4.0 | 7.8 | 7.6 | 20.1 | 57.6 | 0.9 | 1.7 | 4.0 | 5.3 | 58.2 | 76.4 |
| w/o Trioformer | 2.8 | 4.5 | 8.1 | 7.9 | 11.7 | 55.2 | 1.2 | 2.0 | 4.5 | 5.6 | 54.2 | 77.4 |
| 4 Iteration | 2.6 | 4.0 | 7.8 | 7.5 | 14.9 | 58.9 | 1.1 | 1.8 | 4.5 | 5.5 | 52.6 | 77.9 |
| w/o Intra | 2.4 | 4.0 | 7.8 | 7.6 | 17.6 | 56.4 | 0.9 | 1.7 | 4.5 | 5.6 | 55.6 | 78.3 |
| w/o P2Rank | 2.0 | 4.2 | 7.8 | 7.6 | 24.2 | 56.4 | 0.8 | 1.7 | 4.1 | 5.6 | 54.8 | 79.1 |

Table S2: Ablation Study of E3Bind. For discussions, see Section 4.2.

## E.2  IMPACT OF POST OPTIMIZATION

To investigate the impact on time cost and performance of different post-optimization methods, we plug each model into each post-optimization method and log the performance in Table S3. In practice, fast point cloud fitting is faster but yields worse results. This justifies the choice of gradient descent as the post-optimization method for E3Bind. Note that for the two benchmarked post-optimization methods, our model consistently outperforms EquiBind and TankBind.

| | | LIGAND RMSD | | | | | | CENTROID DISTANCE | | | | | |
| | | Percentiles ↓ | | | | % Below ↑ | | Percentiles ↓ | | | | % Below ↑ | |
| Post Optimization | Method | 25% | 50% | 75% | Mean | 2 Å | 5 Å | 25% | 50% | 75% | Mean | 2 Å | 5 Å |
|---|---|---|---|---|---|---|---|---|---|---|---|---|---|
| Fast Point Cloud Fitting (0.01 s) | EquiBind | 3.8 | 6.2 | 10.3 | 8.2 | 5.5 | 39.1 | 1.3 | 2.6 | 7.4 | 5.6 | 40.0 | 67.5 |
| | TankBind | 3.1 | 5.8 | 9.8 | 8.4 | 9.1 | 44.6 | 1.3 | 3.0 | 8.2 | 6.6 | 40.5 | 66.4 |
| | E3Bind | **2.3** | **4.1** | **7.8** | **7.4** | **19.3** | **57.8** | **0.8** | **1.5** | **4.0** | **5.1** | **59.0** | **78.8** |
| Gradient Descent (0.44 s) | EquiBind | 3.5 | 5.8 | 10.2 | 8.0 | 4.4 | 42.7 | 1.3 | 2.7 | 7.0 | 5.5 | 40.8 | 68.6 |
| | TankBind | 2.6 | 4.2 | **7.6** | 7.8 | 17.6 | 57.8 | **0.8** | 1.7 | 4.3 | 5.9 | 55.0 | 77.8 |
| | E3Bind | **2.1** | **3.8** | 7.8 | **7.2** | **23.4** | **60.0** | **0.8** | **1.5** | **4.0** | **5.1** | **60.0** | **78.8** |

Table S3: Performance and run time on a 16-CPU machine of deep-learning models paired with post-optimization methods on the flexible blind self-docking task. Fast point cloud fitting (Stärk et al., 2022) changes torsion angles of the initial pose to best match the generated pose by performing maximum likelihood estimates of von Mises distributions. Gradient descent (Lu et al., 2022) minimizes the weighted sum of the protein-ligand distance error w.r.t. the predicted distance map and the intra-ligand distance error w.r.t. the reference distances.

## F    SENSITIVITY TO INITIALIZATION

E3Bind model's iterative refinement start from some initial structure at the functional block. The initial structure is generated by RDKit and placed in the functional block with random rotation and translation. Here we investigate the influence of initialization for the model's final performance. We generate 30 pose predictions for each protein-ligand complex with different translation and rotation. As shown in Figure S3, 80% of the test set examples have their standard deviation of ligand RMSD within 0.196 Å and standard deviation of centroid distance within 0.188Å. The result indicates E3Bind's low sensitivity to ligand pose initialization and the strong robustness of its iterative refinement process.

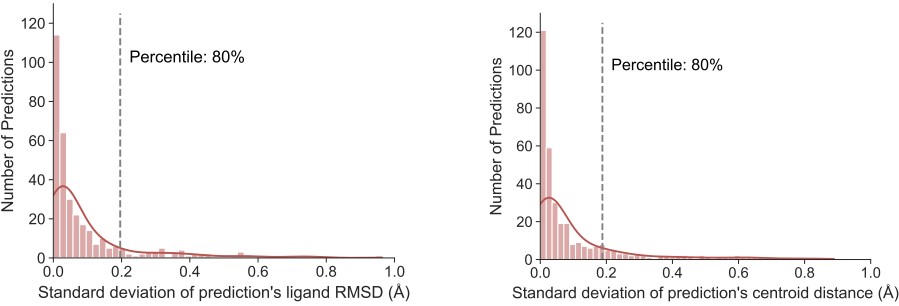

Figure S3: Experiment result sensitivity for different initialization

## G    INFERENCE SPEED

The inference speed of different methods is shown in Table S4. Here we report the average time cost in seconds for a single prediction. As shown in Section G, we can arbitrarily compose deep learning methods and post-optimization methods. Therefore, here we compare the inference speed of their uncorrected versions. For TankBind and E3Bind, P2Rank segmentation time ($\sim 0.15$ s in parallel) is included. For TankBind, the time for distance-to-coordinate optimization ($\sim 0.44$ s in parallel) is also included. As shown in Table S4, the inference speed of E3Bind exceeds traditional methods by a large margin.

| Method | Avg. Sec. 16-CPU | Avg. Sec. GPU |
|---|---|---|
| QVINA-W | 49 | - |
| GNINA | 247 | 146 |
| SMINA | 146 | - |
| GLIDE | 1405* | - |
| VINA | 205 | - |
| EquiBind-U | 0.14 | 0.03 |
| TankBind-U | 1.2 | 0.87 |
| E3Bind-U | 2.2 | 0.44 |

Table S4: Inference Speed. Numbers for score-based methods are taken from the EquiBind paper (Stärk et al., 2022). *The GLIDE only uses a single thread since the multi-threaded version requires a separate license.

## H ADDITIONAL TRAJECTORIES AND CASE STUDIES

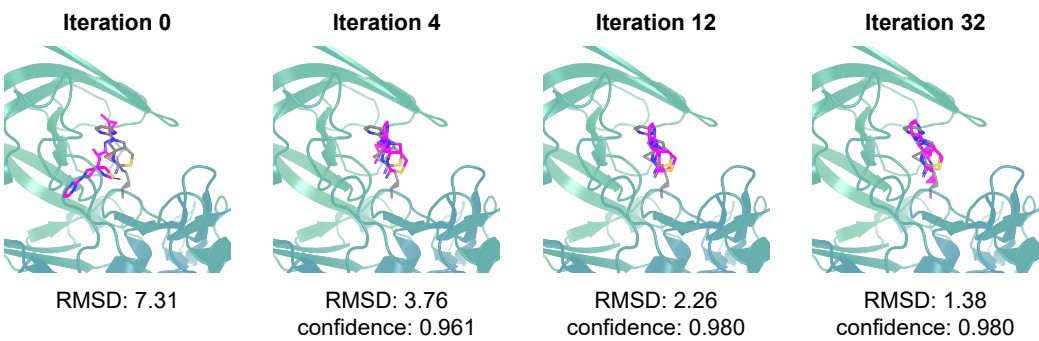

| Iteration 0 | Iteration 4 | Iteration 12 | Iteration 32 |
|---|---|---|---|
| RMSD: 7.31 | RMSD: 3.76 confidence: 0.961 | RMSD: 2.26 confidence: 0.980 | RMSD: 1.38 confidence: 0.980 |

Figure S4: E3Bind coordinate refinement trajectory for ligand in PDB 6UWP. In each figure, The ground-truth docked ligand pose is shown in gray and predicted structures shown in magenta. RMSD and model confidence are written below the figures. E3Bind achieves an RMSD of 1.38 Å after 32 iterations.

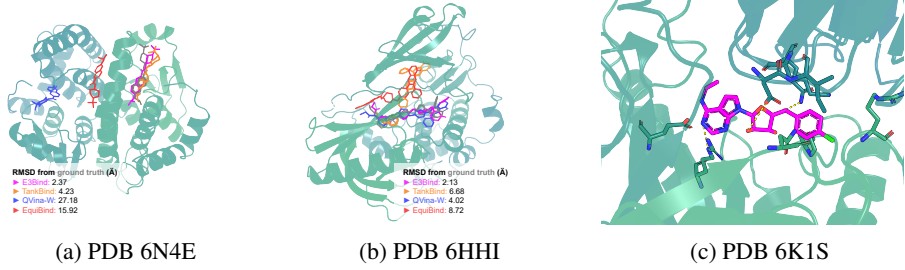

(a) PDB 6N4E      (b) PDB 6HHI      (c) PDB 6K1S

Figure S5: Additional case studies. (a, b) Predicted pose from E3Bind (magenta), TankBind (orange), EquiBind (red) and QVina-W (blue) are placed together with the target protein. RMSD from ground truth ligand pose (grey) are shown on the figure. (a) E3Bind correctly identifies the binding site from the large protein, while EquiBind and QVina-W are off-site. The predicted structure of E3Bind has a low RMSD of 2.37 Å. (b) In this relatively small protein, E3Bind produces ligand pose with the lowest RMSD. (c) An example of a potential binding pocket predicted with high self-confidence by E3Bind. The predicted ligand pose is shown in magenta and protein (including backbone cartoons and sidechain sticks) are shown in green. Contact between the predicted pose and ths protein are highlighted in yellow.

# I EXAMINING THE VALIDITY OF GENERATED STRUCTURES

## I.1 BOND LENGTH DISTRIBUTION

Here we examine the validity of our generated structures by comparing the bond length distribution plots between the predicted and ground truth structures. We can see from Figure S6 that E3Bind outputs have very similar bond distributions compared to the ground truth.

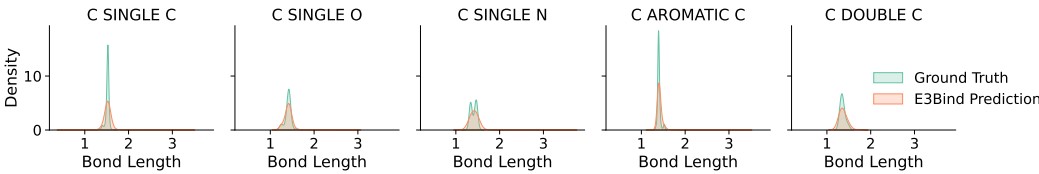

Figure S6: Distribution of bond lengths in E3Bind-predicted (orange) and ground truth (green) ligand structures, grouped by bond types.

## I.2 STERIC CLASH PROBLEM

Steric clash is a common problem in docking, where two non-bonding atoms show an unnatural overlap in the predicted structure. While traditional docking methods avoid generating steric clashes by incorporating a repulsion term in the scoring function, deep learning models have yet to solve this issue.

In our experiments, we find that E3Bind produces very few steric clashes [3] (**3.3%** of the test set contain at least one steric clash), while EquiBind (Stärk et al., 2022) are more likely to produce severe steric clashes (**21%** of the test set have clashes) as shown in S7b. This is mainly because EquiBind docks the ligand into protein by key-point matching in one shot manner, which prohibits further ligand conformational changes to fit in the protein pocket. On the other hand, E3Bind iteratively refines the ligand's position, orientation and conformation, which means it could correct its previous errors, resulting in much fewer steric clash problems.

It is also observed that our self-confidence prediction module plays an important role in selecting structures with fewer steric clashes. Without the confidence module, E3Bind with only 4 iterations has a **13.8%** change of generating a pose with steric clashes. When we apply the confidence module for filtering, the fraction drops to **6.6%**. Subsequent iterations of E3Bind further decrease the fraction to **3.3%**.

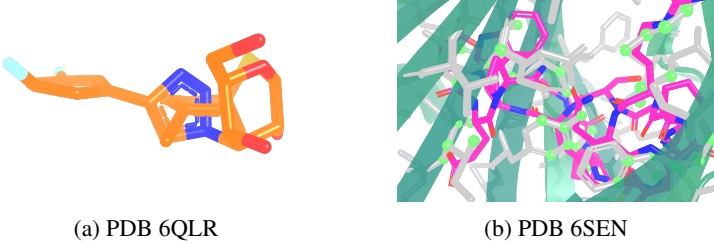

(a) PDB 6QLR                    (b) PDB 6SEN

Figure S7: Illustration of two types of steric clashes. (a) TankBind predicted pose containing a knotted structure, which is physically impossible. The error is due to an unrealizable distance map predicted by TankBind. (b) EquiBind predicted pose (in magenta) with its protein context. The protein atoms within 1.5 Å of the ligand (*i.e.* having severe clashes with the ligand) are highlighted with a lime sphere. Stick models of the corresponding residues are shown in gray. The protein cartoon is shown in dark green.

---

[3] A steric clash is defined by a pair of heavy atoms with a distance of less than 0.4 Å following (Ramachandran et al., 2011).

