# OpenReview forum: "E3Bind: An End-to-End Equivariant Network for Protein-Ligand Docking"
_ICLR.cc/2023/Conference — ICLR 2023 poster_

### Official Review · Reviewer_GTTE · 2022-10-18

**Confidence:** 4
**Correctness:** 3
**Technical Novelty And Significance:** 2
**Empirical Novelty And Significance:** 3
**Recommendation:** 5

**Clarity, Quality, Novelty And Reproducibility:**

Clarity: Overall, the writing of the paper feels rushed and could use from further rewriting and language editing, as it reads very clunkily (and informal) at times.

Novelty: The proposed model deserves merit, but it is somewhat incremental, especially when considering other very similar approaches such as EquiBind. Many of the base ideas of iteratively updating coordinates using the EGNN model are borrowed from the aforementioned study, albeit it is used differently.

Reproducibility: The authors do not provide any accompanying code with this submission, complicating potential replication of the results.

**Strength And Weaknesses:**

Strengths:

* Noticeable improvement over well-established benchmark

Weaknesses:
* Motivation for the architecture somewhat lacking
* Incremental work
* Manuscript feels rushed and reads clunkily at times


**Summary Of The Paper:**

In the proposed work, the authors present E3bind, an fully-equivariant and differentiable approach for protein-ligand docking. Similar works have recently become available in the literature, the claimed advantage of the presented one being that it iteratively docks the ligand into a (possibly unknown) protein binding pocket.

Most of the questions/issues for the authors are presented in the review summary section.

**Summary Of The Review:**

Overall, while this is very solid work and deserves to be published, I believe it needs further work, as many of the parts feel rushed. I would have also enjoyed if the authors had provided code to back up the claims supported in the study and for overall reproducibility, as this is a field that is very rapidly advancing in the last couple of years.

* In the introduction, the authors make a comparison to other methods that only output a "scalar" score. It should be noted that this is also the case for classical docking algorithms, whose scoring function also outputs a scalar for a specific pose. The amount of information provided is somewhat similar - I believe the comparison is not well defined here.
* The claim on the introduction that the performance of Equibind is not strong enough needs further motivation, so as to claim that rapid generation of binding poses remains an unsolved problem.
* On the Related Works section, there is a claim about Equibind failinf to beat popular docking baselines without finetuning which needs to be further justified/motivated.
* Overall, figure captions on the manuscript could be heavily improved upon (within reasonable constraints of the page limit) as they are not descriptive at all.
* In general, I think the reader would have benefited from further details about the Trioformer architecture, as it is mentioned without it being properly presented. Is this naming convention used in previous works?
* In Section 3.1, the authors mention that they use only the carbon alpha atoms to represent the protein in the model. Was this choice made for computational efficiency issues, or did the authors compared against  a full-atom representation?
* Why are the message passing networks of the protein (GVP-GNN) and ligand (GIN) chosen differently? There does not seem to be any motivation for this choice.
* In section 3.3, the authors claim that the iterative refinement method allows the ligand to sense the local environment compared to the one-shot approaches, but no motivation behind this intuition is provided.
* On the self-confidence prediction idea. The plot on Appendix Section C.2 seems to indicate that these scores are not very well aligned with ligand RMSD, as they tend to concentrate on the right-hand side of the distribution (most of them over 0.9). Could the authors consider providing an appropriate scaling as well as checking the pearson correlation between the score and rmsd?
* In Section 3.4, is the loss only computed on predicted vs. experimental ligand coordinates?
* In Section 3.4 (inference process), the authors mention that they use a binding site algorithm to first segment the protein into functional blocks. Are the other methods also making use of this step? Could this potentially bias the improvements shown in the results? How computationally expensive is this segmentation step - is it taken into account when computing the numbers provided in Appendix Section G? The authors also mention in this section that their model does not require binding affinity for training but this is usually never the case for docking tasks, so I do not understand why this is explicitly mentioned.
* How is the ligand initialization done? The authors claim that it is randomly initialized in each segmented block before docking. Does this result in possible clashes with the protein target?
* In terms of the performance on flexible self docking scenarios, it should be noted that that many standard docking pipelines (and not only GLIDE, as mentioned in the paper) actually outperform the DL-based alternatives.
* On the "benefit of iterative refinement" section. How does the self-confidence score behave under non-successfully docked examples?

Other notes:
* Page 2:
* Page 2: "it's" -> "its"
* In the Geometry-aware pair updates section, why is the word "neighboring" under quotes?
* Equations are missing commas and periods when appropriate.
* Equation 10, the loss is missing equation number.
* Is the \sigma in the self-confidence prediction section a sigmoid function?

---

> ### Author Response · Authors · 2022-11-19
> **Response to Reviewer GTTE (3/3)**
>
> **Q10: In Section 3.4 (inference process), the authors mention that they use a binding site algorithm to first segment the protein into functional blocks. Are the other methods also making use of this step? Could this potentially bias the improvements shown in the results? How computationally expensive is this segmentation step - is it taken into account when computing the numbers provided in Appendix Section G? The authors also mention in this section that their model does not require binding affinity for training but this is usually never the case for docking tasks, so I do not understand why this is explicitly mentioned.**
>
> **(1) segmentation method issues**
>
> **A:** Thanks for pointing this out! The P2Rank segmentation is used previously in TankBind. We used the exact same P2Rank version (v2.3) for a fair comparison. While P2Rank can help segment protein into high quality functional blocks, our ablation study (Table S2) has shown that the proposed E3Bind can achieve comparable performance even if P2Rank is replaced by random segmentation. It’s mainly due to the effectiveness of the self-confidence module.
>
> As for the segmentation time you pointed, it is taken into account in Section G, which is **~ 0.15 s / protein** in parallel. We have also added a note to Section G clarifying the situation. Thank you for the kind reminder!
>
> **(2) “not requiring binding affinity for training”**
>
> **A:** It’s indeed that requiring the binding affinity is usually not the case in the problem of computational docking. We mentioned this because some previous deep learning based model like TankBind utilize the additional affinity label to train their docking model as an auxiliary task. It may have some misleading parts, so we polished the text.
>
> ______
>
> **Q11: How is the ligand initialization done? The authors claim that it is randomly initialized in each segmented block before docking. Does this result in possible clashes with the protein target?**
>
> **A:** Yes, for flexible blind self-docking, the ligand is initialized from an RDKit-generated structure and randomly placed inside the block. The initial pose could clash with the protein target. However, this is not an issue as the clash will gradually be fixed during iterative refinement of the ligand pose. Further details can be found in our new Appendix I.2.
>
> _______
>
> **Q12: Possible issues for performance compared with standard docking pipelines in flexible self docking scenarios**
>
> **A:** Thank you for the comment! We believe that **for the general test set, E3Bind achieves superior performance compared with standard docking pipelines.** For the unseen protein test set, it’s true that DL-based alternatives including E3Bind fail to beat standard docking pipelines in some metrics (<2Å). It’s probably due to the data scarcity problem in deep learning. There are only **~3500 unique** proteins in the training set. It only covers a very small part of the whole protein family, which result in the data scarcity problem. A potential solution for this issue is to utilize some pretrained protein structure encoder or language model. We leave it as a future work to combine with them.
>
> _______
>
> **Q13:  How does the self-confidence score behave under non-successfully docked examples?**
>
> **A:** Thanks for pointing this out. We added the distribution of self-confidence score in both successful examples (RMSD <5Å) and unsuccessful examples (RMSD >= 5Å) to Appendix C.2. It’s observed that these two distributions have high divergence. For successful docking examples (RMSD < 5A), the average confidence score is **0.907**, while for unsuccessful docking examples, the average confidence score is **0.612**.
>
> |  | Success |Non-Success |
> | -------- | ------- | ------- |
> | Average Score| 0.907   |0.612 |
>
> _______
>
> **Issues in other notes**
>
> Thanks a lot for pointing these. We have polished data and fix the typo to make them more clear based on your valuable suggestions. $\sigma$ is represented as a sigmoid function to map the confidence score to [0,1]. Thanks again for your time and review!

---

> ### Author Response · Authors · 2022-11-19
> **Response to Reviewer GTTE (2/3)**
>
> **Q5: Why residue-level representation?**
>
> **A:** The choice was made for computational efficiency issues, because the time and space complexity of the Trioformer are O(*n*p2*n*c + *n*p*n*c2). We also note that residue-level representation is prevalent among methods on protein, such as AlphaFold2[1], ESM-1b[2], ESM-IF[3], EquiBind[4] and so on.
>
> **References**
>
> [1]. Cramer, Patrick. "AlphaFold2 and the future of structural biology." *Nature Structural & Molecular Biology* 28.9 (2021): 704-705.
>
> [2]. Rao, Roshan M., et al. "Msa transformer." *International Conference on Machine Learning*. PMLR, 2021.
>
> [3]. Hsu, Chloe, et al. "Learning inverse folding from millions of predicted structures." *bioRxiv* (2022).
>
> [4]. Stärk, Hannes, et al. "Equibind: Geometric deep learning for drug binding structure prediction." *International Conference on Machine Learning*. PMLR, 2022.
>
> ______
>
> **Q6: Why are the message passing networks of the protein (GVP-GNN) and ligand (GIN) chosen differently?**
>
> **A:** These two encoders are widely used separately for protein and ligand in previous work [1]. We believe it’s well suited for the problem of docking to fetch the initial feature because
>
> 1. Protein 3D structure is assumed to be known in current docking settings. GVP has been proven to be a powerful 3D structure encoder beating many other counterparts.
> 2. Ligand 3D structure is usually unknown in docking. While there are many popular structure encoders for small molecules (such as SphereNet, Dimenet++ and SEGNN), these powerful encoders all require the full 3D structure of the input. These constraints drive us to use the 2D graph encoder GIN to encode ligand graphs following previous work. In practice, we find that it achieves relatively good performance.
>
> **References**
>
> [1]. Lu, Wei, et al. "TANKBind: Trigonometry-Aware Neural NetworKs for Drug-Protein Binding Structure Prediction." *bioRxiv* (2022).
>
> [2]. Liu, Yi, et al. "Spherical message passing for 3d molecular graphs." International Conference on Learning Representations. 2021.
>
> [3]. Klicpera, Johannes, et al. "Fast and uncertainty-aware directional message passing for non-equilibrium molecules." arXiv preprint arXiv:2011.14115 (2020).
>
> [4]. Brandstetter, Johannes, et al. "Geometric and physical quantities improve e (3) equivariant message passing." International Conference on Learning Representations. 2022.
> ________
>
>
>
> **Q7: In section 3.3, the authors claim that the iterative refinement method allows the ligand to sense the local environment compared to the one-shot approaches, but no motivation behind this intuition is provided.**
>
> **A:** Iterative refinement is previously used by AlphaFold2 and has proven its power due to the parameter reusing and ability to better sense the local environment.
>
> Intuitively, **one-shot approach docks the ligand pose by a single-step prediction. So they have zero knowledge about the spatial relationship between protein and ligand when making predictions.** In an iterative refinement framework, instead, the ligand gradually finds its pose. As the ligand gradually moves to the true position, the spatial relationship between protein and ligand becomes more accurate. Correspondingly, our model can better construct the spatial graph and perform spatial message passing between protein and ligand, sensing which protein residue should interact with which ligand atom. The more expressive message passing in turn drives the model to find a better pose… So generally, **iterative refinement method allows the model to construct an inter-graph based on the current protein-ligand spatial relationship and to sense the local environment for the ligand.**
>
> A typical one-shot approach is EquiBind, where the model docks the ligand to protein in one-shot matter with keypoint alignment. In experiment, we find that **such one-shot methods usually cause steric clashes, where our model significantly reduces the steric clash partly due to its better ability to sense the local environment.**
>
> _______
>
> **Q8:  Could the authors consider providing an appropriate scaling as well as checking the pearson correlation between the score and rmsd?**
>
> Thanks for your valuable suggestions! We have rescaled the plot and calculated the correlation metrics for score and negative RMSD in the appendix based on your suggestions. The Pearson correlation coefficient is **0.641** and the Spearman correlation coefficient is **0.783**. It’s worth noting that the goal of the self-confidence prediction module is to select the final pose. The value of Spearman correlation coefficient shows that our self-confidence prediction module’s selection is reasonable in most cases.
>
> | Spearman | Pearson |
> | -------- | ------- |
> | 0.783    | 0.641   |
>
> ______
>
> **Q9: is the loss only computed on predicted vs. experimental ligand coordinates?**
>
> Yes. We have revised the text so that the notations are more easily understood.

---

> ### Author Response · Authors · 2022-11-19
> **Response to Reviewer GTTE (1/3)**
>
> Thanks a lot for your insightful suggestions! We have devised some statements to make our manuscript more informative and added some experiments based on your valuable comments. We have also attached our code and added a reproducibility statement.
>
> **Q0: Comparision between the proposed model and previous work EquiBind.**
>
> **A**: Thank you for your comment! Actually, our method is very different from EquiBind. The central idea of EquiBind is to separate the docking process into two phases: (1) iterative conformation adjustment for the unbound ligand via IEGMN, an EGNN-GAT hybrid, and (2) keypoint alignment where the adjusted ligand is globally roto-translated into the pocket.
>
> While the first stage shares the similar idea of iterative refinement, it’s worth noting that in EquiBind (i). there is no encoder like Trioformer to ensure the basic geometric constraints when adjusting the ligand conformation; (ii). the ligand’s position and orientation are predicted by keypoint alignment in a one-shot manner, so there is no rescue if after this roto-translation the ligand cannot fit into the pocket. The above issues have caused the problem of severe steric clashes (See revised Appendix I): about **21% of predicted structures in EquiBind suffer from steric clashes.**
>
> Instead, **there is only 3% steric clash cases in our prediction**. Our model addresses the above issues by (a). proposing a geometric constraint-aware encoder Trioformer to better sense the distance constraint (b). unifying the docking as a single iterative refinement process instead of two stages so that ligand can simultaneously refine its conformation and translation/rotation. We have polished the text accordingly to convey the idea more clearly.
>
> ______
>
>
>
> **Q1: The comparison of docking algorithms and other methods that only output a "scalar" scores may not be well defined in the introduction**
>
> **A1:** Thank you for your suggestion! We were thinking that for docking, in addition to the score, plausible docked poses are also generated which gives more information and increased interpretability. But indeed, it is not very meaningful to compare these two tasks like this. We have revised the text accordingly to avoid confusion.
>
> _________
>
>
>
> **Q2: Claim about EquiBind's performance needs further justification.**
>
> **A:** Thank you for your comments! What we wanted to express is “EquiBind failing to beat popular docking baselines without combining it with traditional docking software”. From table 1 of our paper, we can see that only 5.5% of EquiBind-predicted ligand pose has an RMSD < 2Å, while for QVina-W the proportion is 20.9%. Only when combined with SMINA can the performance of EquiBind increase to 24.6%, surpassing QVina-W, according to table 1 of the EquiBind paper. Therefore we claim that "EquiBind fails to beat popular
> docking baselines on its own, stressing the importance of increasing model expressiveness". We speculate that the use of the term “fine-tuning its prediction” was causing some confusion, and we have revised the text accordingly.
>
> | Model            | % Below 2Å ↑ | % Below 5Å ↑ |
> | ---------------- | ------------ | ------------ |
> | EquiBind         | 5.5          | 39.1         |
> | EquiBind + SMINA | 24.6         | 46.4         |
> | QVina-W          | 20.9         | 40.2         |
>
>
>
> __________
>
>
>
> **Q3: Figure caption on the manuscript needs improvement**
>
> **A:** Thank you for the important advice for our paper! We have rewritten the figure captions to make them more clear and more informative.
>
> ________
>
> **Q4: In general, I think the reader would have benefited from further details about the Trioformer architecture, as it is mentioned without it being properly presented. Is this naming convention used in previous works?**
>
> **A:** Thank you for the suggestion. We added more details of the Trioformer in the main text and the appendix. The Trioformer naming is due to: (1) the embedding “trio” that it updates, namely the protein embedding, ligand embedding and pair embedding; (2) the triangle attention it used to enforce geometric consistency.

---

### Official Review · Reviewer_5skX · 2022-10-24

**Confidence:** 4
**Correctness:** 3
**Technical Novelty And Significance:** 3
**Empirical Novelty And Significance:** 3
**Recommendation:** 6

**Clarity, Quality, Novelty And Reproducibility:**

Clarity: Good. The paper is overall well-written and organized.

Quality: Excellent. The performed experiments are extensive.

Novelty: Medium. The proposed method is based on existing methods. The key contribution is to integrate Alphafold with TankBind and address the problem of conformation refinement.

Reproducibility: Bad. No code is provided.


**Strength And Weaknesses:**

Strengths:
1. The proposed method addresses the two-stage optimization problem of previous work by performing coordinate iteration through a context-aware equivariant module, enabling end-to-end training.
2. The paper is overall well-written and easy to follow.
3. The proposed method shows superior empirical performance over existing baselines.

Weaknesses and questions:

1. The technical contributions should be further clarified. The architecture of this model is similar to the combination of Alphafold and TankBind. Although it solves the separated coordinate optimization problem of TankBind, the additional modules are mainly borrowed from Alphafold. The authors should elaborate more on their own contributions.
2. It seems that pair embedding z_ij does not change during the iterative coordinate update. Is this on purpose or a compromise made because of computation cost?
3. The authors could comment on why the performance on unseen protein is still worse than GLIDE as measured by RMSD <2Å. Does this indicate that the model has poor generalization ability? Also, the performance gain over TankBind is not significant.
4. There is a recent work DiffDock that can also iteratively generate poses. The authors may need to consider differentiation from DiffDock (https://arxiv.org/abs/2210.01776) or state the superiority of the proposed method.
5. Binding affinity prediction is a key subtask for protein-ligand interaction prediction. Could this work perform affinity prediction and how?



**Summary Of The Paper:**

This work tackles the problem of protein-ligand binding structure prediction. The authors propose an E(3) equivariant network to iteratively update the poses. Alphafold-inspired feature encoders are introduced to capture the information of proteins and ligands, respectively. The whole architecture is trained in an end2end fashion. The model could achieve state-of-the-art performance in both binding position prediction and binding affinity prediction tasks.


**Summary Of The Review:**

Generally, I think it's a good paper. But the clarity of novelty should be improved and some of the results should be further explained. Also, no code has been released. I vote for a borderline acceptance.

---

> ### Author Response · Authors · 2022-11-19
> **Response to reviewer 5skX (part 2/2)**
>
> **Q5**: Could this work perform affinity prediction and how?
>
> **A**: Binding affinity prediction is indeed an important task. Although our model is not designed for this task, we can still do it by attaching a binding affinity prediction head to the end of the E3Bind decoder. The prediction head has the same architecture as the self-confidence prediction head but is trained with an MSE loss on binding affinity. We did a simple trial without fine-tuning hyperparameters. The results (below) show that our model achieves comparable results with the previous SOTA model.
>
>
> | Model|RMSE     | Pearson | Spearman | MAE   |
> | -------- | ------- | -------- | ----- | ----- |
> | TransCPI | 1.741   | 0.576    | 0.540  | 1.404 |
> | MONN     | 1.438   | 0.624    | 0.589 | 1.143 |
> | PIGNet   | 2.640   | 0.511    | 0.489 | 2.110  |
> | IGN      | 1.433   | 0.698    | 0.641 | 1.169 |
> | HOLOPROT | 1.546   | 0.602    | 0.571 | 1.208 |
> | STAMPDPI | 1.658   | 0.545    | 0.411 | 1.325 |
> | TankBind | 1.371   | 0.718    | 0.665 | 1.111 |
> | E3Bind   | 1.379   | 0.687    | 0.652 | 1.100 |

---

> ### Author Response · Authors · 2022-11-19
> **Response to reviewer 5skX (part 1/2)**
>
> Many thanks for your time and review!
>
> **Q1**: The architecture of this model is similar to the combination of AlphaFold and TankBind. The authors should elaborate more on their own contributions.
>
> **A:** Thank you for the suggestion! The main contribution of this work is: (1) formulating docking as an iterative structural refinement task, where the output pose of a decoder block is fed as the initial pose of the next one, allowing the model to **dynamically sense the local context and fix potential errors**, (2) proposing an end-to-end equivariant network for docking, so that the **training objective is better aligned with the task at hand** and (3) proposing a novel way to **injecting geometry constraints into the pair embeddings**. Indeed, various components of our model are based on previous work, but combining them and adapting them for docking is a nontrivial task. For example, we could not trivially apply AlphaFold2 for docking, because (1) the fixed protein structure must be considered, (2) the ligand atoms do not have a local frame as in protein residues and (3) the EvoFormer module does not take into account the geometric clues implicit in docking. E3Bind addresses the above issues in an elegant way.
>
> _____
>
>
> **Q2**: Pair embedding z_ij does not change during the iterative coordinate update. Is this on purpose or a compromise made because of computation cost?
>
> **A:** Good questions! There are mainly two reasons for such a design. First, **reducing computational complexity**. The geometry-aware Trioformer relies on triangle attention which has cubic $O(n_\text{p}^2n_\text{l} + n_\text{l}^2n_\text{p})$ computational complexity. It’s thus expensive to merge it into each decoder layer. Second, **preserving the geometry constraints**. We treat the pair embeddings as constraint-aware guidance that helps the decoder generate geometry-consistent poses. It is undesirable for the Trioformer-injected distance constraints to be diluted. Given the above consideration, we fix the pair embeddings during decoding.
>
> _____
>
> **Q3**: Why the performance on unseen protein is still worse than GLIDE? Does this indicate that the model has poor generalization ability?
>
> **A**: Thank you for pointing this out! Docking for unseen protein is indeed a challenge for deep learning based models, as they often require a large number of high-quality data to achieve good generalization ability. The number of unique proteins in the training set is only ~3500, which covers a very small portion of the whole protein family. In our humble opinion, this is what leads to the poor generalization ability of current deep learning based models. A potential solution is to incorporate pretrained protein structure encoders or language models. These models are trained on large-scale standardized datasets and thus could enhance the model’s generalization ability. We believe it is a promising future direction.
>
> _____
>
> **Q4**: Differentiation from DiffDock / state the superiority of the proposed method?
>
> **A**: Thank you for the comment and reference. We think DiffDock is a very interesting work that uses an SE(3)-equivariant neural network to generate the translation, rotation and conformation (defined by torsion angles of the rotatable bonds) of the docked structure. Its results also demonstrate the effectiveness of iterative structural refinement and the usefulness of the self-confidence model.
>
> The performance gain is likely due to: (1) DiffDock **narrows down the degrees of freedom** of the output space from $3n$ to $m+6$, where $n$ is the number of ligand atoms and $m$ is the number of rotatable bonds in the ligand, making the model easier to train; (2) it contains **much more parameters** and adopts **pretrained ESM2 features**.
>
> Compared to DiffDock, the strength of E3Bind lies in (1) **cheaper training and inference**, as it does not rely on the tensor product of steerable vectors, and (2) **less inductive bias and potentially better adaptability**. The $m+6$ output space, though very elegant, depends on the notion of rotatable bond in molecules and is nontrivial to transfer to another domain, e.g. constrained point cloud generation.

---

### Official Review · Reviewer_eHD3 · 2022-10-25

**Confidence:** 4
**Correctness:** 4
**Technical Novelty And Significance:** 3
**Empirical Novelty And Significance:** 3
**Recommendation:** 6

**Clarity, Quality, Novelty And Reproducibility:**

**Clarity.** The paper is written in a clear way and easy to follow. The model description is supported by figures that show the model overview as well as atom pairing. Most of the experimental and implementation details are provided.

**Quality.** The experimental results are convincing. The main results are supplemented by additional ablation experiments and plots. The selection of models in the benchmark is comprehensive and up-to-date.

**Novelty.** The proposed model is another attempt at creating a neural-networks-based molecular docking algorithm. This is a relatively new branch of ML research that can accelerate drug discovery. The method demonstrates a different view on this problem as the atom position prediction is here formulated as a multi-step refinement procedure. It uses several novel methods for solving the system equivariance, including the proposed Trioformer architecture.

**Reproducibility.** The code is not included in the supplementary material. The model description is clear, and the supplementary materials contain the algorithm pseudocode and implementation details, so it should be possible to reimplement the model. However, all implementation details can be difficult to recover without the source code because of the number of architecture elements.

**Strength And Weaknesses:**

Strengths:
- The selection of graph neural networks for encoding the ligands and binding pockets is reasonable.
- Trioformer is proposed to tackle the problem of preserving the geometry constraints when modeling atom pair embeddings.
- I really like the self-confidence estimator that is trained along with the docking objective. Figure 2 clearly shows the benefits of this predictor.
- The case study shows the usefulness of the proposed method in a real-life scenario.
- An ablation study was conducted, where different components of the architecture were removed.
- A quantitative evaluation of the improvement over multiple refinement steps is presented, as well as a qualitative refinement trajectory in the supplementary materials.

Weaknesses:
- Is Trioformer a new architecture that is introduced in this paper? This should be emphasized more, and the differences between Trioformer and former architectures (Evoformer, triangle attention) should be accentuated in my opinion.
- In the problem definition paragraph, the naming of blind docking problems seems a little confusing to me. Is “re-docking” and “self-docking” used synonymously? If so, maybe the names “rigid blind self-docking” and “flexible blind self-docking” would be better in this context?
- The model does not have any formal constraints on the validity of the generated 3D structure. Atoms can move freely, and, although the shown structures look realistic, there is no guarantee that the resulting compounds are not knotted or do not include very long bonds. Can you provide a plot showing a distribution of bond lengths? Another interesting plot would be depiction of the number of invalid structures (steric clashes) as the number of refinement steps increases.
- The code was not included in the submission, so it is difficult to assess the reproducibility of the experiments.

Questions:
- How can you output rigid structures in the rigid docking experiment? According to the footnote on page 5, you start with the same conformation as the docked ligand structure, but do you change this conformation via per-atom position updates?
- Only to confirm, is gradient detached from the RMSD term when training the self-confidence predictor? How do you select beta to achieve the best tradeoff between the docking performance and top pose selection?
- In the experimental results, what corrections are applied to each model? Is the set of post-processing methods the same across different methods?

Minor points:
- Typos, e.g.
“we achieves SOTA in most metrics” -> “we achieve SOTA in most metrics”
“Figure ??” -> “Figure 5”
“prediction’s” -> “predictions”
“uncorrcted” -> “uncorrected”
- Some references should be fixed, e.g. referencing Section B in Section 3.2 (it should be indicated that the appendix is referenced), referencing “E” in Section 4.2, and referencing Figures 5 and 7 that are not included in the main text body.
- Section G of the Appendix should be referenced in Section 4.1, where there is a statement about superior inference speed - currently this feels like an unsupported claim.


**Summary Of The Paper:**

The paper describes a new method for molecular docking that uses equivariant neural networks. The positions of ligand atoms are refined in each step, and the per-atom translation is computed by DecoderLayer that uses equivariant graph convolution layers. The protein and ligand are encoded with graph neural networks and protein-ligand interactions are modeled by Trioformer, a transformer updating atom pair information based on triplets of atoms from both the protein and ligand. The model can be trained in the end-to-end manner. The presented results are competitive compared to the recent neural docking algorithms and the classical ones.

**Summary Of The Review:**

Based on the above comments, I am leaning towards the acceptance of this paper if all the issues are addressed by the authors.

---

> ### Author Response · Authors · 2022-11-18
> **Response to Reviewer eHD3 (2/2)**
>
> **Q3: In the experimental results, what corrections are applied to each model? Is the set of post-processing methods the same across different methods?**
>
> **A:** Thanks for your questions! The reported results in the paper are obtained with their original implementation and post optimization method. Specifically, EquiBind uses fast point cloud fitting which changes torsion angles of the initial pose to best match the generated pose by performing maximum likelihood estimates of von Mises distributions. TankBind uses gradient descent which minimizes the weighted sum of the protein-ligand distance error w.r.t. the predicted distance map and the intra-ligand distance error w.r.t. the reference distances. E3Bind uses the latter for post optimization. We have revised the paper to clarify this (Section 4.1 baseline).
>
> To investigate the impact on time cost and performance of different post optimization methods, we plug each model into each post-optimization method and log the performance. In practice, **fast point cloud fitting is faster but yields slightly worse results**. This justifies the choice of gradient descent as the post optimization method for E3Bind. Note that **for the two benchmarked post-optimization methods, our model consistently outperforms EquiBind and TankBind**. These results and analysis have been added to **Appendix Section E.2.**

---

> > ### Comment · Reviewer_eHD3 · 2022-11-23
> > **Thank you for your response!**
> >
> > Thank you for the reply and for addressing my comments. I appreciate the time spent on running more experiments included in the supplementary materials and including the code for better reproducibility.

---

> > > ### Author Response · Authors · 2022-12-14
> > > **Thank you!**
> > >
> > > Dear reviewer eHD3,
> > >
> > > Thanks a lot for your time reading our response and for appreciating our revision.

---

> ### Author Response · Authors · 2022-11-18
> **Response to Reviewer eHD3 (1/2)**
>
> Thank you very much for your review! We have revised our paper following your valuable comments and suggestions. The detailed responses are as follows:
>
> **W1: Is Trioformer a new architecture that is introduced in this paper?**
>
> **A:**  Thank you for the great suggestion! We have revised the paper to lay more emphasis on this. The key difference between Trioformer and Evoformer is that **Trioformer incorporates implicit geometric constraints in docking (see $t^{(h)}_{jk^\prime}$ in Eq. 1)**. Since docking is a special structure prediction problem where basic geometric constraints (protein-protein spatial distance, ligand-ligand bond length, ..) should be obeyed. encoding these constraints is essential in model design.
>
> _________
>
> **W2. Possible name ambiguity in problem definition**
>
> **A:** Thanks for pointing this out! “Re-docking” is linked with the rigid-ligand setting because we only need to restore the position and orientation of the docked ligand, and “self-docking” is linked with the flexible-ligand setting because we are docking the ligand back to its cognate protein. So in this logic “re-docking” is a form of “self-docking” where the ligand conformation is given. The difference is minor though, and these two docking settings are usually contrasted with *cross-docking* (docking a ligand to a non-cognate receptor) rather than each other. We use the current terminology following EquiBind and TankBind, as consistency with related works is a desideratum when we draft our paper.
>
> ________
>
> **W3: Validity of generated structure**
>
> **A:** Thanks for your constructive suggestions! We have added a section in the appendix to discuss the validity of the generated structure.
>
> For our current decoder design, it’s true that it allows the atom to move freely in space. It’s likely to generate “deformed” point clouds. So we combine it with a proper post optimization method to guarantee the output structure’s validity. Common post optimization methods include fast point cloud fitting and gradient descent, which we discuss in Appendix E.2.
>
> We have also added the bond length distribution plot for different bond types following your valuable suggestions. It’s observed that **the bond length of the generated structure shows a very similar distribution with ground truth.**
>
>  Also thanks a lot for pointing out the possible steric clash problem! We added a section in Appendix I to discuss it. It’s interesting to find that the proposed **E3Bind model predicts much fewer steric clash cases (around 3%) compared with previous deep learning models**. We attribute the success to
>
> 1. The confidence module has helped filter out some steric clash cases.
> 2. The iterative refinement framework helps the ligand gradually fit the pocket, avoiding potential steric clashes.
>
> _______
>
> **W4: Code reproducibility**
>
> **A:** Thank you for the important suggestion. We have now included the code in the submission and added a reproducibility statement.
>
> ___________
>
> **Q1: How can you output rigid structures in the rigid docking experiment? Do you change this conformation via per-atom position updates?**
>
> **A:** Yes, E3Bind changes the conformation via per-atom position updates as we focus on the more practical flexible docking setting. Theoretically, the conformation could deviate from the given pose after several coordinate update iterations, but with proper post-optimization we could still arrive at the given conformation in most cases. E.g., the point cloud fitting algorithm in EquiBind guarantees that the final pose is the same as the given pose.
>
> ________
>
> **Q2: Is gradient detached from the RMSD term when training the self-confidence predictor? How do you select beta to achieve the best tradeoff between the docking performance and top pose selection?**
>
> **A:** Yes, we detach the RMSD gradient when training the self-confidence predictor (added this to the main text). We select beta so that the scale of the two losses are roughly the same. In our experiments, we did not observe significant negative impact on docking performance when jointly training the confidence model.

---

### Official Review · Reviewer_ofea · 2022-10-31

**Confidence:** 4
**Correctness:** 4
**Technical Novelty And Significance:** 3
**Empirical Novelty And Significance:** 3
**Recommendation:** 6

**Clarity, Quality, Novelty And Reproducibility:**

The paper was generally an easy read and the figures and tables straight-forward to interpret.

**Strength And Weaknesses:**

strengths:
+ The encoder-decoder architecture for ligand-protein binding can be considered novel although comprised of modules that have been proposed in the literature before, save for the following point.
+ The distance-based attention mechanism to update the pair embeddings seems to somewhat respect the 3D Euclidean geometry of the individual protein and the individual ligand in an interaction.
+ The general setup has taken many inspirations from recent works on protein structure modeling and protein binding including alphafold.
+ From (although limited) ablation studies it seems at least some of the additions are generally successful in improving the docking performance.
+ The results show clear improvements over recent baselines on protein-ligand binding.

possible improvements:
- The proposed method is comprised of many learnable components. It would be informative to compare the capacity of the model in some form with prior similar works, at least the more recent ones EquiBind and TankBind. Then, it would be important to empirically verify that it is not the increased capacity that explains the improved performance.

- An important ablation study seems to be missing for the distance-based attention mechanism in updating the pair embeddings which seems to be the main technical novelty of the work apart from the general constellation of the approach.

**Summary Of The Paper:**

The paper proposes an end-to-end encoder-decoder approach for ligand-protein binding (blind docking) where the ligand can conform while the protein structure is assumed fixed. In the encoder part, the individual atomic representation of ligand and residue-based representation of ligand is learnt through graph networks which in turn leads to an embedding for pairs of atoms and residues and are all update within a transformer. The decoder iteratively processes the embeddings to output Euclidean coordinates for the ligand atoms where it respects the SE(3) equivariance of rotation and translations of the ligand and protein.

**Summary Of The Review:**

The empirical results are relatively strong, the work has the minimal required novelty in the general design as well as the attention mechanism. On the other hand, a conclusive message on the performance can benefit from the above-mentioned additional experiments. I lean towards accepting the paper.

---

> ### Author Response · Authors · 2022-11-18
> **Response to Reviewer ofea**
>
> Many thanks for your helpful and constructive comments!
>
> **Q1: Comparison with previous deep learning based work**
>
> **A**: Thanks for your constructive suggestions! We compare the model capacity with EquiBind and TankBind according to your suggestions. It’s worth noting the parameters of all three deep learning based models are relatively small **(<5M)**, while E3Bind has a slightly larger size due to the self-confidence prediction module and decoder layers. We tried increasing the capacity of TankBind and EquiBind (increasing the number of layers and hidden dims to be comparable with E3Bind). The parameter number of newly implemented EquiBind and TankBind is shown in the table. However, we find that they achieve roughly the same performance as their original implementation. Thus, **we believe that the increase of model capacity is not the determining factor for the performance.**
>
> | Model    | Model Capacity (Original Implementation) | Model Capacity (New Implementation) |
> | -------- | ---------------------------------------- | ----------------------------------- |
> | EquiBind | 1.38M                                    | **3.74M**                           |
> | TankBind | 1.85M                                    | **3.83M**                           |
> | E3Bind   | **3.57M**                                | -                                   |
>
>
>
> _______________
>
> **Q2: Important ablation study for distance-based attention mechanism**
>
> **A**: Thank you for raising this important issue! We have added this ablation study to Appendix Section E.1. The results show that without the geometry-constrained attention bias, the fraction of good poses (with RMSD < 2 Å) degraded from **23.4%** to **20.1%**, highlighting **the benefit of incorporating distance constraints in attentive pair update modules.**

---

### Author Response · Authors · 2022-11-19
**Response to all the reviewers**

We would like to thank you for your time and effort to review our paper. Your kind suggestions and thorough comments provide valuable insights for us to increase the quality of our paper. We have revised our manuscript following your suggestions. Specifically, we
1. included additional ablation study showing the benefit of incorporating geometric constraints in Trioformer and the impact of post-optimization methods,
2. added a section in the appendix discussing the validity of generated structures, by examining the bond length distribution and evaluating steric clashes,
3. polished writing & rewritten figure and table captions for better clarity and presentation.

We also attached code in the supplementary material to promote reproducibility.

---

### Decision · Program_Chairs · 2023-01-20

**Decision:**

Accept: poster

**Justification For Why Not Higher Score:**

See the above-mentioned weaknesses.

**Justification For Why Not Lower Score:**

See the above-mentioned strengths.

**Metareview: Summary, Strengths And Weaknesses:**

The paper describes an end-to-end encoder-decoder approach for ligand-protein binding where the ligand can conform while the protein structure is assumed fixed. In the encoder part, the atomic representation of the ligand and residue representation of the ligand are learned by a GNN that returns embeddings for pairs of atoms and residues and are updated within a transformer. The decoder iteratively processes the embeddings to output Euclidean coordinates for the ligand atoms where it respects the SE(3) equivariance of rotation and translations of the ligand and protein.

**Strengths:**
* Reviewers appreciated that experiments are extensive and noted considerable improvement over well-established benchmarks.
* Trioformer is proposed to tackle the problem of preserving the geometry constraints when modeling atom pair embeddings.
* Reviewers appreciated the self-confidence estimator is trained along with the docking objective and that the benefits of this predictor are demonstrated through a real-world case study.


**Weaknesses:**
* The new method has many learnable components, which reviewers pointed out and requested additional experiments. In response, the authors compared the model capacity with recent EquiBind and TankBind, following reviewers' suggestions. Further, the discussion revealed that the parameters of all three deep learning based models are relatively small (<5M) and that E3Bind has a slightly larger size due to the self-confidence prediction module and decoder layers.
* Reviewers also asked for an ablation study of the distance-based attention mechanism in updating the pair embeddings, which seems to be the main technical novelty of the work apart from the general constellation of the approach. That is an important issue, and the authors added this ablation study to the appendix.


Overall, it is clear the authors addressed all key concerns raised by reviewers. Reviewers appreciated new experiments as well as the authors' additional efforts to improve reproducibility.

**Note From Pc:**

if the above contains the word "oral" or "spotlight" please see: "oral" presentation means -> notable-top-5% and "spotlight" means -> notable-top-25%. As stated in our emails, we are disassociating presentation type from AC recommendations